# Antecedents of the responsible acquisition of computers behaviour: Integrating the theory of planned behaviour with the value-belief-norm theory and the habits variable

**W. H. Loo**[1☯¤a]**, Paul H. P. Yeow**[2‡¤b]**, Yuen Yee Yen**[ID][3*¤c]

**1** Sunway University Business School, Department of Marketing, Sunway University, Selangor, Malaysia, **2** School of Business and Management, RMIT University, Vietnam, Ho Chi Minh City, Vietnam, **3** Faculty of Business, Multimedia University, Melaka, Melaka, Malaysia

☯ These authors contributed equally to this work.
¤a Current address: Sunway University, Jalan Universiti, Bandar Sunway, Selangor Darul Ehsan, Malaysia
¤b Current address: Business Innovation Department, School of Business and Management, RMIT University Vietnam, Ho Chi Minh City, Vietnam
¤c Current address: Faculty of Business, Multimedia University, Melaka, Melaka, Malaysia
‡ PHPY also contributed equally to this work.
* yyyuen@mmu.edu.my

**Data Availability Statement:** All relevant data are within the manuscript and its Supporting Information files.

## Abstract

The responsible behaviour of consumers that purchase green computers is a form of sustainable consumption, as green computers use less energy resulting in less $CO_2$ emissions and the use of fewer toxic metals and materials during their production. The research question is how to encourage such behaviour. Although prior research has provided some answers by investigating the antecedents of the behaviour, it has done so through a piecemeal approach from the angles of the theory of planned behaviour (TPB), the value-belief-norm (VBN) theory, and habits. The present research aims to investigate the antecedents of the responsible acquisition of computers behaviour (RACB) among Malaysian consumers by integrating the TPB and the VBN theory with the habits variable. Hypotheses and a research framework were developed based on these theories and a survey questionnaire was used to collect information on the green computer purchase behaviour of computer owners aged 17 and over in Malaysia. A total of 1,000 usable surveys were completed and structural equation modelling was used to analyse the data collected. The findings reveal that the TPB, the VBN theory, and the habits variable can be integrated to explain RACB, which is formed when biospheric values trigger subjective norms that subsequently result in the formation of habits that lead to intentions of acquiring green computers and RACB. The study's findings show that although personal norms do not affect RACB, subjective norms affect ascriptions of responsibility, personal norms, and RACB. The findings provide insights to policymakers, NGOs, manufacturers, and marketers that can assist them in designing strategies for the effective promotion of RACB.

**Funding:** The author(s) received no specific funding for this work.

**Competing interests:** The authors have declared that no competing interests exist.

## 1. Introduction

Responsible consumption behaviour (RCB) is a pro-environmental behaviour. It is defined as the "acquisition, consumption and disposition of goods, services, time, and ideas by decision-making units without harming the environment or society [1]". Thogersen [2] highlighted that it is worth studying RCB and its effects on environmental quality, particularly at the acquisition stage. This is because a responsible purchasing decision could reduce or eliminate environmental harm at the later stages of the consumption cycle. Joshi and Rahman [3] supported this claim by highlighting that 40% of environmental harm is caused by irresponsible consumption from the purchase of non-sustainable products.

The responsible acquisition of computers behaviour (RACB), on the other hand, involves purchasing a computer without harming the environment. This may involve buying a computer that is compliant with the Electronic Product Environmental Assessment Tool (EPEAT). EPEAT is a standard certified by the GlobalElectronics Council (formerly known as Green Electronics Council) for computers and related products that contain little to no toxic content (such as heavy metals), consume less energy, and are easily upgradeable and recyclable to extend their lifespan [4]. To illustrate the effectiveness of EPEAT on the purchase of computers, the purchase of 868 million between 2006–2014 EPEAT-registered computer products in the US reduced the disposal of 9,740 metric tonnes of toxic computer waste, saved 130.1 million megawatt hours of electricity during product production and use cycles, and reduced greenhouse gas emissions by 24million metric tonnes due to electricity savings [5, 6].

Many studies have been conducted to discover the antecedents of RCB using various theories and factors such as the theory of planned behaviour (TPB), the value-belief-norm (VBN) theory, and the habitual factor [7–10]. The TPB articulates how individuals make reasoned choices and choose alternatives with the greatest benefits against the lowest costs in serving their self-interest, wherein the behaviour is determined by behavioural intention that lies on individual positive assessment on the behaviour (attitude), social pressure adopting the behaviour (subjective norm) and perceived ease of engaging the behaviour (perceived behavioural control) [9, 11–13].

Prior studies substantiated that the TPB could serve as a basic model in explaining RCB encapsulating energy saving [7, 9, 14–16], recycling [17], purchasing organic food [18], green purchase [8, 10, 19] etc. However, the TPB owns its limitation i.e., TPB is a self-interest theory that primarily comprises rational predictors while omitting other relevant RCB contextual factors like moral obligation, habits [20, 21]; thus, affecting the predictive power of the theory. The predictive power of the TPB became higher from 56% to 65% after adding moral norms and self-identity to the original TPB [18, 22]. Additionally, Gao et al. [15] revealed that the explanatory power of TPB raised from 22.6% to 34.9%, and the significant predictors of individual energy-saving behaviour include all the TPB variables (except subjective norm) and the added variables i.e., descriptive norms and personal norms. Wan et al. [17] found two additional variables, i.e., awareness of consequences and moral norms, increased the predictive power of the TPB in predicting recycling intention. Liu et al. [23] revealed the extended TPB model performed well in green purchase intention setting. All these findings shed light for future RCB research that extending TPB is required as the theory fails to consider the moral aspect; thus, it cannot fully explain the behaviour.

RCB is a pro-social behaviour which is not merely predicted by TPB variables, but also could be well-explained by VBN theory, wherein the VBN theory proposes that altruistic behaviour is triggered by the morality factors such as personal norms, ascriptions of responsibility, awareness of consequences, new ecological paradigm, and values. However, VBN theory does not cover the self-interest predictors that are captured by TPB variables. So far, most RCB researches focus mainly on extending TPB with one or a few variables instead of integrating the

whole TPB with the complete VBN theory to provide a holistic view of how the variables of both theories work, notably in high-cost RCB context (e.g. the RACB of present study).The findings of Ates [24] revealed that integrating TPB variables with morality variables accounts for more explained variance in pro-environmental behaviour ($R^2 = 0.488$) compared to the use of individual TPB ($R^2 = 0.464$). Ates [24] highlighted that other important variables (e.g., egoistic value, altruistic value, new ecological paradigm belief) presented in the VBN should be taken into consideration. However, Ates [24] is confined to general RCB, not specific behaviour like green purchase behaviour. Another study by Li et al. [25] combined the TPB, the diffusion of innovation theory (DOI), and the personal norm variable into a conceptual framework. However, the integration of the whole VBN variables were excluded from their study.

Although the previous literature highlighted that habits should be included in studies for a more accurate prediction of repetitive behaviour, this factor had been ignored in the TPB and VBN models [26–28]. In response, extensive research added habits into the TPB model. For instance, Cheung et al. [26] revealed that the explained variances of the TPB increased from 32.5% to 67.4% after incorporating habits into the TPB model to explain wastepaper recycling behaviour. Besides, the impact of TPB variable i.e. SN on VBN factors (e.g. AR, and PN), and habits are under-explored [29].

Moreover, most of the prior RCB studies that combined the TPB and the VBN theory were focused on general pro-environmental behaviour [24, 30] or low-cost, pro-environmental behaviour in the choice of transportation [31, 32] and the purchase of organic products [25]. The present study considers high-cost pro-environmental behaviour in the purchase of green computers that are more expensive than conventional computers.

To address the above research gaps, the objective of the study is to investigate the antecedents of RACB by integrating the TPB and the VBN theory with the habits variable. The outcome of this research will provide new knowledge of how the theories can be integrated to explain RACB. With this understanding, recommendations can be provided to policymakers, NGOs, manufacturers, and marketers to promote sustainable consumption through RACB.

This research was conducted in the developing economy context of Malaysia, as a limited number of studies of RCB [33–36] have been conducted to date in such a context where most of the population is comprised of low- to middle-income groups and the country has a collectivistic culture. Moreover, the Malaysian government is interested in understanding the determinants of RACB as it has included the United Nations Development Program's Sustainable Development Goal 12 (responsible consumption and production) in its national development plan [37]. Additionally, due to the expansion of green products into developing countries like Malaysia, Indonesia, and India, local and international manufacturers and marketers are interested in understanding the antecedents of RCAB in such countries to assist them in penetrating the green computer market [35].

In brief, this study found that the TPB, the VBN theory, and the habits variable can be integrated to explain RACB, and that biospheric values trigger subjective norms, which subsequently result in the formation of habits that lead to intentions and RACB. Although personal norms do not affect RACB, subjective norms are found to affect ascriptions of responsibility, personal norms, and RACB.

## 2. Hypotheses development and conceptual framework

### 2.1 Attitude Towards Responsible Acquisition of Computers Behaviour (ATRACB)

Attitude has been defined "as the enduring positive and negative feeling about some performing targeted behaviour" [38]. Attitude should be measured specifically [39]. Felix and

Braunsberger [40] substantiated that due to the failure of not measuring environmental attitude, the factor did not impact green products purchased. Following Follows and Jobber's [39] suggestion and Felix and Braunsberger [40]'s finding, this study investigates a specific behaviour (RACB); hence, attitude is operationalised specifically as the degree to which a consumer has a favourable or unfavourable evaluation about performing RACB.

Previous research has validated and substantiated that a positive attitude will lead to greater positive behavioural intention in the context of RCB. Harland et al. [41] established this regarding pro-environmental behaviour, such as using unbleached paper and energy-saving light bulbs and turning off faucets. Follow and Jobber [39] did so regarding environmental purchasing behaviour, while Chan and Lau [42] established this for green purchase behaviour. Nguyen et al. [43] found this in the purchase behaviour associated with energy-efficient household appliances, Joshi and Rahman [3] with sustainable purchasing behaviour, and Ates [24] with the purchase of eco-labelled food. Based on these results, it is expected that Malaysian consumers with a positive attitude about RACB will be more likely to engage in such behaviour. Thus, the following hypothesis is postulated:

H1: Attitude towards RACB positively affects an individual's intention of engaging in RACB.

## 2.2 Subjective norms (SN)

The term "subjective norms" refers to "the individual's perception of social pressure to perform the particular behaviour" [38] in the TPB model. In other words, an individual is conscious of whether other people believe that he or she should perform or not perform a behaviour. These "other people" are his or her friends, relatives, family members, peer groups, and other reference groups [44]. This study adopts Ajzen [45]'s operationalised definition.

SN have been found by several studies [35, 44] to have direct and significant effects on behavioural intentions to engage in responsible purchasing. The strength of SN is based on "normative belief" and "motivation to comply". "Normative belief" refers to whether others think that an individual should or should not perform responsible purchasing behaviour, whereas "motivation to comply" refers to an individual's motivation to follow social expectation [35]. Similarly, this study demonstrates that the opinions and expectations of a consumer's referent groups, such as family members, friends, or teachers, will influence their intention of practising RACB. As such, the following hypothesis is formed:

H2a: Subjective Norms positively affect an individual's intention of engaging in RACB.

Further investigation into the effects of SN in determining behaviour in the TPB is required after the significant direct effects of SN on behaviour (rather than intention) were found, particularly in Asian countries with collectivist cultures [3, 46, 47]. This is because individuals act in their own interest and conveniently infer that their behaviour is sensible if other people perceive it to be so [48]. In other words, individuals will purchase green computers if their referents (friends, family, etc.) perceive they should purchase them. Thus, the following hypothesis is offered:

H2b: Subjective Norms positively affect an individual's engagement in RACB.

In line with the TPB, while SN affect an individual's intention, the impact of SN on personal norms (PN) is not presented in the TPB model. PN (an individual's view on what is right or wrong) are rooted in SN. SN provide the standard of behaviour that a social reference group views as appropriate in a specific context. As an individual subsequently internalises these standards, they provide the basis of his or her PN [30, 49]. Brekke et al. [50], glass-recycling behaviour among Norwegian households led to the finding that feelings of PN become strong if people consider that recycling is morally right among their friends and family. Similarly, if

RACB is socially approved, then the behaviour will serve as the basis for an individual's norm (i.e., that performing RACB is his or her moral obligation. Hence, the following hypothesis is formed:

H2c: Subjective Norms positively affect an individual's Personal Norms in RACB.

Numerous studies have mentioned that SN shape beliefs in addition to behaviour [16, 51, 52]. Ru et al.'s [16] experiment discovered friends sharing medical information among themselves and shaping one another's beliefs about the cost of catching the flu and the perception of the risk of infection if not vaccinated. While this supported the notion that peer influence affects an individual's beliefs, the study was confined to the health context. Consistent with this, the present study also postulates that a consumer's belief about their responsibility for $CO_2$ emissions and hazardous waste pollution from not purchasing computers responsibly (known as ascription of responsibility, or AR) stems from friends, family, etc. (i.e., SN), and sharing information about RACB practices. Therefore, the following hypothesis is proposed:

H2d: Subjective Norms positively affect an individual's Ascription of Responsibility in RACB.

Researchers also proposed that SN might affect an individual's habits, such as those related to eating, buying, or alcohol consumption [53]. Mukama [53] reasoned that individuals are willing to change/maintain their habit of alcohol consumption owing to SN (i.e., disapproval or approval of such habits and the motivation to comply with the expectations of others). In other words, individuals will continue to engage in a habit if the habit is sanctioned by their social group. Conversely, they will discontinue engaging in a habit if it is not sanctioned by their social group. A study conducted by Kandel et al. [54] mentioned that children who are firm friends share similar attitudes and engage in similar habits, such as alcohol consumption, due to social pressure. Similarly, it is posited that if the RACB of purchasing green computers or purchasing computers only when necessary is supported by referents, individuals will maintain their existing habit of practising RACB or shift their habit of purchasing non-green computers to purchasing green ones. Therefore, the following hypothesis is proposed:

H2e: Subjective Norms positively affect an individual's habits in practising RACB.

## 2.3 Perceived behavioural control (PBC)

PBC refers to "people's perceptions of the ease or difficulty of performing the behaviour of interest" [38]. Specifically, PBC reflects two dimensions: (a) an individual's external conditions that may augment or moderate his or her ability to adopt a certain behaviour, and (b) an individual's perceived ability to exercise the behaviour. In the context of computer-purchase decisions, PBC is conceptualised in this study according to Ajzen [38].

In line with the TPB, PBC predicts specific responsible purchasing behaviour directly and indirectly through intentions [45]. Prior studies have found that PBC influences pro-environmental responsible purchasing behaviour intention and actual responsible purchasing behaviour for activities ranging from recycling to the purchase of green household appliances [55]. Researchers like Ates [56] and Yuriev et al. [21] identified that consumers with a high degree of control over the barriers or discomfort associated with pro-environmental behaviour will involve themselves in such behaviour. In a similar vein, Malaysian consumers who perceive the existence of conducive external conditions like the availability of eco-labels, advertisements related to RACB, green computers, and so forth will spur their RACB either directly or indirectly via behavioural intention. Therefore, the following hypotheses are proposed:

H3a: Perceived Behavioural Control positively affects an individual's intention of engaging in RACB.

H3b: Perceived Behavioural Control positively affects an individual's engagement in RACB.

## 2.4 Responsible acquisition of computers behavioural intention (RACBI)

Behavioural intention is defined as "the likelihood of an individual performing the behaviour in the future" [38]. RACBI was defined by Ajzen [45] as "consumer perception about his/her likelihood of using a green product responsibly in the future". An intention indicates a person's efforts and willingness to try and perform the behaviour [38]. The present study conceptualises RACBI as a consumer's effort to engage in acquiring computers responsibly after a cognitive deliberative process. If an individual has a strong intention to engage in the behaviour, he or she is likely to perform it [45]. Follows and Jobber [39] claimed that the omission of intentions resulted in low correlations between antecedent variables and behaviour. This is supported by Balderjahn's [31]finding that the relationship between attitude towards pollution and purchase behaviour is not significant. To further validate the relationship between behavioural intention and actual behaviour, the following hypothesis is offered:

H4: Behavioural intention positively affects an individual's engagement in RACB.

## 2.5 Values and the new ecological paradigm (NEP)

According to the VBN theory, values are "unique beliefs that lead to actions and judgements". They consist of two functions, which are "patterns (conduct guides)" and "motivation (the expression of efforts to realise a value)" [57]. Stern and colleagues [58–60] proposed the existence of three different value orientations: egoistic (values focusing on maximising individual outcomes), altruistic (values reflecting concern for the welfare of others), and biospheric (values emphasising the environment and the biosphere). In addition, the new ecological paradigm (NEP) in the VBN chain refers to a set of general beliefs about the relationship between humans and nature [61]. The NEP emphasises beliefs in the limits of growth, the necessity of balancing economic growth with environmental protection, and the need for the preservation of nature [62]. In this study, the terms "new ecological paradigm" and "environmental concern" are interchangeable. This study conceptualises and operationalises the definitions of the egoistic, altruistic, and biospheric values and NEP based on the VBN theory.

Congruent with the specification of the VBN model, basic values will shape an individual's NEP [63]. The empirical findings of Stern et al. [58] provide evidence that the values people hold are indicative of how they see themselves in relation to the environment. The findings of these studies illustrate that individuals who hold weak egoistic values and strong altruistic and biospheric values are more likely to accept the NEP. Hence, the following hypotheses are tested in this study:

H5a: Egoistic values negatively affect Environmental Concerns.

H5b: Altruistic values positively affect Environmental Concerns.

H5c: Biospheric values positively affect Environmental Concerns.

In addition, biospheric values are prominent, compared to altruistic values, when explaining beliefs regarding environmental behaviour [58–60]. The empirical findings of Ng and Cheung [64] substantiate this. Their findings show that the placing of value on the environment by children is vital to the development of their pro-environmental beliefs and behavioural intention to perform recycling and conservation. Nonetheless, studies of the impacts of biospheric values on SN are underexplored [24, 65, 66]. Theoretically, individuals with biospheric values may enhance their "normative beliefs" and "motivation to comply" with others' expectations to practise RACB, which is congruent with their biospheric values. In this study, only biospheric values is chosen, as it is closely related to pro-environmental behaviour compared with egoistic and altruistic values.H6: Biospheric values positively affect an individual's Subjective Norms in practising RACB.

## 2.6 Awareness of consequences (AC)

Awareness of Consequences (AC) is defined as an individual's belief that environmental conditions threaten individual values [67]. Stern et al. [68] and Garling et al. [69] measured AC focused on general environmental conditions instead of environmental conditions associated with specific behaviour. Mo et al. [10] conceptualised that environmental concern should be related to the demand for protecting environment. Other researchers, such as Steg et al. [61], highlighted that the VBN theory may be enhanced by tuning AC to specific behaviour [62]. Hence, this study conceptualises AC related to not practising RACB. The consequences include negative environmental problems caused by greenhouse gas emissions, the depletion of resources, and so forth, that threaten what an individual value; the valued object could be oneself, others, or the biosphere.

The VBN chain delineates that the NEP shapes an individual's AC. People with a stronger concern for the environment will be more aware of the impact of their actions on themselves, others, and the environment than those who are less concerned about environmental issues, as reflected in Stern's [61] findings. Also, Ates [56] discovered that the NEP plays a critical role in forming an individual's AC. This study posits that the Malaysian consumer's environmental concern will determine his or her specific belief that not practising RACB will pose environmental threats to themselves, others, or the biosphere. Hence, the following hypotheses are tested in this study:

H7a: An individual's Environmental Concerns negatively affect his or her Awareness of Consequences to themselves from not practising responsible acquisition of computers behaviour.

H7b: An individual's Environmental Concerns positively affect his or her Awareness of Consequences to others from not practising responsible acquisition of computers behaviour.

H7c: An individual's Environmental Concerns positively affect his or her Awareness of Consequences to the biosphere from not practising responsible acquisition of computers behaviour.

## 2.7 Ascription of responsibility (AR)

Ascription of Responsibility (AR) refers to an individual's belief that he or she bears significant [59, 61] responsibility for the consequences of their behaviour. This study conceptualises AR as an individual's belief that they are responsible for the consequences of not practising RACB.

Stern et al. [58] stated that an awareness of adverse consequences (AC) will lead to AR. Individuals with a high awareness of AC are presumed to be aware of the extensive and specific consequences of actions. As a result, they adopt the perspective of those who will be affected when weighing decisions. Since AC is positively related to AR, individuals who are more aware of the adverse consequences of their actions will feel more responsible to reduce the threats resulting from them [68, 70]. As such, the following hypotheses are tested in this study:

H8a: An individual's Awareness of Consequences to oneself positively affects his or her Ascription of Responsibility for acquiring computers responsibly.

H8b: An individual's Awareness of Consequences to others positively affects his or her Ascription of Responsibility for acquiring computers responsibly.

H8c: An individual's Awareness of Consequences to the biosphere positively affects his or her Ascription of Responsibility for acquiring computers responsibly.

## 2.8 Personal norms (PN)

Samarasinghe [71] defined PN as self-expectations based on internalised values, which may also be expressed as feelings of a personal obligation to engage in certain behaviour. Ajzen [45]

found that personal norms were influenced by social expectation; thus, did not distinguish the two factors. However, Lind et al. [62] distinguished personal norms from social norms. According to Lind et al. [62], the expectations, sanctions, and obligations tied to personal norms are anchored in oneself, but social norms are anchored in a social group [50]. Valle et al. [72] further explained that personal norms reflect the beliefs held by an individual about how they should behave. For example, an individual will experience a keen sense of pride if he or she follows these norms. In contrast, individuals undergo a feeling of guilt if their personal norms are violated. In line with this general idea, this study conceptualises personal norms as an individual's feelings that practising RACB is his or her personal obligation.

Stern [61] highlighted that AR will lead to the activation of PN, resulting in a moral obligation to act. This is supported by the previous studies of Garling et al. [69], Steg et al. [68], and Lind et al. [62]. To reflect the VBN causal chain, this study predicts that individuals will feel that practising RACB might be a personal moral obligation if they feel that they are responsible for the consequences of not engaging in it. The preceding arguments lead to the following hypothesis:

H9: Ascription of Responsibility positively affects Personal Norms for the responsible acquisition of computers.

Pro-environmental behaviour is morally right behaviour that maximises environmental benefits. It is driven by PN (i.e., feelings of a moral obligation to undertake pro-environmental action), which is consistent with the VBN model. This is evidenced in past studies where personal norms were identified as determinants of behaviour in environmental domains like energy conservation, recycling, and pro-environmental buying [30, 73]. Although Stern et al. [59] and other researchers such as Steg et al. [68] revealed the strong positive impact of PN on behaviour in the context of low-cost pro-environmental behaviour, the effect of PN on high-cost pro-environmental behaviour remains unknown. This study postulates that a consumer's adoption of RACB lies in the intensity of his or her feelings of a moral obligation to practise RACB in the context of high-cost pro-environmental behaviour. Hence, the following hypothesis is tested in this study:

H10: An individual's Personal Norms positively affect the responsible acquisition of computers.

## 2.9 Habits

The variable "habits" is used in the extended TPB model [21]. Habits have been described as "an automatic link between a goal and a specific behaviour and, as opposed to more controlled behaviour, demands very little attention and subsequent elaboration" [73]. Alibeli [74] defined habit as "the automatic performance of behaviour triggered by context cues". According to Thogersen and Olander [2], habits will evolve based on three requirements. First, "the behaviour needs to be repeated". Ouellette and Wood [75] clarified that behaviour that is repeated annually or once in one to two years is known as infrequent habitual behaviour, while behaviour performed daily and regularly, such as driving or recycling, is frequent habitual behaviour. Behaviour that is not repeated cannot be considered habitual behaviour. Second, "the behaviour must take place in stable surroundings" so the habit can be developed. Researchers have reasoned that habits are learnt actions in response to constant situations. For instance, an individual may have a habit of reading a particular magazine at a hairdressing salon but they never read it elsewhere. The instigation cue or constant situation is "at the salon". Third, "rewarding consequences must be available". This means that habits are developed by the systematic experience of consequences that reward the behaviour. In the case of RACB, the practice will become a habitual behaviour after it is performed repeatedly at least once every one to

two years in response to specific cues (e.g., the purchase of a new computer in response to the cue of the replacement of an existing computer due to excessive wear and tear), and the practice provides a reward through the achievement of certain green goals or end states involving the environment, saving energy, etc. While most of the existing studies have equated habits with automatic responses in the context of frequent behaviour, behaviour performed repeatedly in the past might not elicit automatic responses if it was performed infrequently. Hence, the term "habits" in this study is conceptualised as the extent to which people repeatedly and responsibly (rather than automatically) acquire computers.

Habits are conceptually related to green-consumer behaviour [30, 76, 77]. Yurie et al. 21] reviewed articles relating to habits and concluded that not all behaviour requires rational thinking; therefore, habits predict actual behaviour directly. Nevertheless, Verplanken et al. [28, 78] and Ouellette and Wood (62) proposed that performing habitual behaviour infrequently (i.e., once a year or every two to three years) might still involve reasoning process of forming the intention of performing the behaviour. In line with their propositions, it is predicted that consumers' habits related to RACB will determine their likelihood in engaging themselves to purchase green computers. Thus, the following hypothesis is tested in this study:

H11: An individual's habits positively affect his or her Responsible Acquisition of Computer Behavioural Intention.

## 2.10 Conceptual framework

Fig 1 shows this study's conceptual framework and the relationships among the variables in the proposed 11 hypotheses.

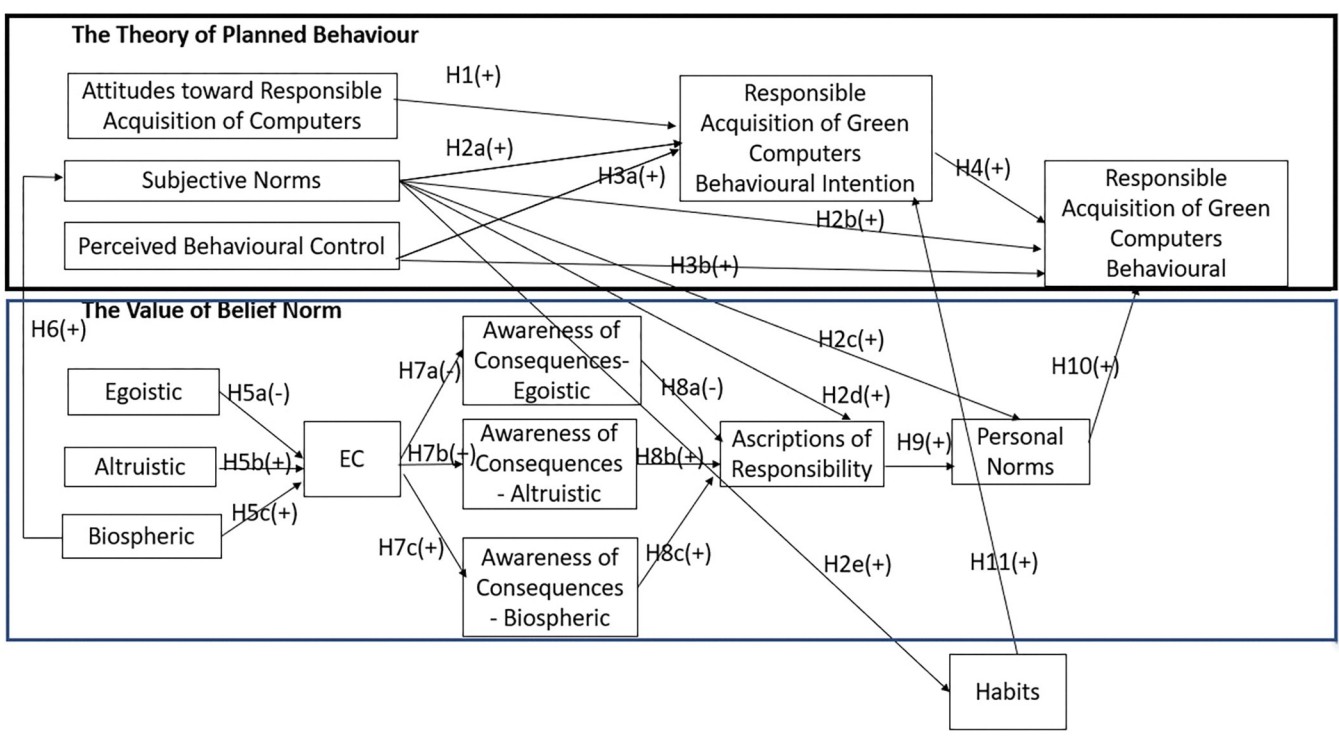

**Fig 1. Conceptual framework.**

## 3. Method

### 3.1 Measures

The measures for all factors in this study are shown in the S1 Appendix. Appendix A in S1 Appendix shows the measures for the dependent variable. RACB. The measures consist of three items adapted from Ajzen [38] and Murugesan [79]. Appendix B in S1 Appendix shows the measures for all TPB factors. They consist of 15 items adapted from Ajzen [38] and Murugesan [79]. Appendix C in S1 Appendix shows the measures for all VBN factors. They consist of 33 items adapted from Stern et al. [59] and Steg et al. [68]. Appendix D in S1 Appendix shows the measures for the habits factor. They consist of three items adapted from Venkatesh et al.[80]. All items were measured using five-point Likert scales with values that include strongly disagree (1), disagree (2), neutral (3), agree (4), and strongly agree (5). While the items were adopted from past studies, some of their wording was changed to suit the present context. The validity and reliability of the constructs were tested using confirmatory factor analysis (CFA) and Cronbach's alpha. In the final model, the composite reliability (CR) value for all factors is above 0.70, which indicates acceptable internal consistency [81]. The average variance extracted (AVE) value for all factors is above 0.5 indicating acceptable convergent validity. Table 3 shows that the $\sqrt{\text{AVE}}$ values of each construct (see the diagonal values) are greater than the correlation values in the same row. This indicates high discriminant validity. Multicollinearity was not an issue in the model since all the correlation values are below 0.8 [82]. Cronbach's alpha, a reliability test statistic, was calculated for all factors and all values were above the threshold of 0.7 [81].

The items measuring the TPB and VBN variables were assessed using a five-point Likert scale that was validated in previous studies by Sawitri et al. [67] and Fu et al. [83]. Revilla et al. [84]'s findings evidenced that a five-point scale is the best scale, in terms of data quality and response rate, for agree and disagree questions.

### 3.2 Ethics clearance

To comply with the regulations of Monash University, ethical clearance had been obtained from the Monash University Human Research Ethics Committee (MUHREC) prior to the distribution of the questionnaire.

We used informed and verbal consent. An explanatory/ethic statement was presented in the front page of the questionnaire (see Appendix D in S1 Appendix). The explanatory statement informed the participants about purpose of the research, the voluntary participation, the anonymity of respondents, the confidentiality of the data collected, and the contact details. The verbal consent was sought. The process was the data collector requests the participants to read the explanatory statement, then gets their verbal consent to participate in the research before letting them answer the questionnaire.

### 3.3 Pilot study

The original questionnaires were pretested for content validation by 10 experts consisting of researchers, lecturers, and professors. Based on their feedback, amendments were made to the questionnaire content, format, terms, ease of completion, and comprehensibility.

A pilot study was then conducted and 100 questionnaires were distributed in public places. Cronbach's alpha, a reliability test statistic, was calculated for all factors and all values were above the threshold of 0.7 [82].

### 3.4 Sampling method

Purposive non-random sampling was used to obtain data from respondents who met the criteria of owning a computer and being 17 years old or above. Individual consumers were chosen

as the unit of analysis because 40% of environmental degradation is attributed to individual consumers' irresponsible consumption of non-sustainable products [3]. Non-random sampling was used because the population frame of all computer owners in Malaysia was not available. We invited only those 17 and above because those below this age range would not normally possess the purchasing power to buy a computer since they have not started working yet [85]. The minimum sample size for this study based on Kline's (2005) ratio of 5:1 for an ideal SEM model is 255 responses (calculated from 51 observed variables x 5). A total of 1000 responses was collected, which is well above the minimum sample size.

### 3.5 Data collection

A total of 1,050 self-administered survey questionnaires were distributed in public places such as shopping complexes, bus stations, parks, etc. throughout all 14 states and federal territories in Malaysia. Before getting their informed consent to participate in the research, the respondents were asked if they had purchased a computer and if they were 17 years old and above. Out of the total questionnaires collected, 50 were excluded from the final analysis because respondents had not purchased computers and/or were less than 17 years of age. As a result, the final sample size was 1000 responses.

### 3.6 Data analysis

The Structural equation modelling (SEM) is known as the most appropriate technique for testing complex model with a large number of constructs [81]. SEM tests a large number of independent and dependent constructs statistically in a simultaneous analysis to develop a structural model, which is a plausible representation of the relations between all variables [81]. SEM is a comprehensive approach of data analysis whereas traditional multivariate procedures are incapable of either assessing and correcting measurement error. SEM provides explicit estimates of the measurement error through confirmatory hypothesis-testing approach [81]. Many researchers like Gao and Bai [7]; Kiatkawsin and Han [86]; Vincent et al. [87]; Nguyen [88] etc. validated the causality among the independent and dependent constructs in the integrative research framework via SEM technique.

The Analysis of Moment Structures (AMOS) software was used to analyse the data. AMOS is a covariance-based SEM technique that can integrate paths, perform factor analyses, and examine multiple relationships in a simultaneous manner [89].

As suggested by Anderson and Gerbing [90], the two-step approach was used in this study. In the first step, the measurement model was analysed for adequacy, and confirmatory factor analysis (CFA) was used to test the validity and reliability of the measurement model. The second step involved testing the structural model and hypotheses through assessing the path coefficients for each hypothesised relationship.

### 3.7 Respondents' demographics

As shown in Table 1, 46.1 per cent of respondents were female and 53.9 per cent were male. More than three-quarters of the respondents were aged 17 to 32, and more than 80% had an income below USD998 (RM4,000). Most respondents had a diploma-level education. The respondents' demographics reflect the ratio of the Malaysian population's characteristic [46].

## 4.0 Results

### 4.1 Confirmatory factor analysis

The first step was to test the measurement model's goodness of fit. The summary of the final measurement model is presented in Table 2. We deleted 14 indicators with low loadings

**Table 1. Respondents' demographic information.**

| | | Frequency | Percent |
|---|---|---|---|
| Gender | Male | 539 | 53.9 |
| | Female | 461 | 46.1 |
| Age | 17–32 years old | 766 | 76.6 |
| | 33–47 years old | 198 | 19.8 |
| | 48–66 years old | 29 | 2.9 |
| | 67 years old | 7 | 0.7 |
| Monthly income | RM0-RM2,000 | 576 | 57.6 |
| | RM2,001-RM4,000 | 243 | 24.3 |
| | RM4,001-RM6,000 | 122 | 12.2 |
| | RM6,001-RM8,000 | 31 | 3.1 |
| | Over RM8,000 | 28 | 2.8 |
| Highest education level | Primary | 14 | 1.4 |
| | Secondary | 214 | 21.4 |
| | STPM | 50 | 5.0 |
| | Diploma | 361 | 36.1 |
| | Degree | 280 | 28.0 |
| | Master's degree | 61 | 6.1 |
| | PhD | 20 | 2.0 |

(<0.70). These were AC6, Altru 1, Altru3, AR2, ATGCPB2, E2, E3, E5, EC2, EC3, PBC2, PBC4, PN2, and SN2. We did not delete GCPB3 because the value was only slightly below 0.70 (0.631). According to Byrne (2005), this item may be retained if it does not violate other indices. We found that retaining this item did not violate the composite reliability (CR) and convergent validity indices. In the final model, the CR value for all factors is above 0.70, which shows acceptable internal consistency [81]. The average variance extracted (AVE) value for all factors is above 0.5, which shows acceptable convergent validity.

## 4.2 Discriminant validity

Table 3 shows that the $\sqrt{\text{AVE}}$ value (see the diagonal values) of each construct is greater than the correlation values in the same row. This indicates high discriminant validity. There are no multicollinearity issues in the model since the values of all correlations are below 0.8 [82].

## 4.3 structural equation modelling

The second step was to test the structural model proposed by this study. The SEM analysis shows the chi-square/df of the initial integrated model is 5.805, which exceeds the requirement of 3; thus, model trimming was performed [81]. The non-significant paths, from ATRACB to RACBI, SN to RACBI, PBC to RACBI, PBC to RACB, Altru to EC, EC to ACego, and PN to RACB were deleted. The path from Ego to EC was also deleted since the beta value is less than 0.10; the effect is trivial [63]. After deleting these paths, the ACego to AR path became non-significant so it was also deleted. Since the indicator of ACaltru showed a negative variance, the EC to ACaltru and ACaltru to AR paths were deleted as recommended by McDonald (1985). After deleting this path, the revised model is shown in Table 4 and Fig 2. The revised model achieved satisfactory goodness of fit, with an $X^2$/DF of 2.906 (< = 3), CFI of 0.987 (> = 0.90), TLI of 0.984 (> = 0.90), GFI of 0.945 (> = 0.90), and RMSEA of 0.044 (< = 0.08).

**Table 2. Summary of the final measurement model.**

| Construct | Items | Loadings | Sig. | AVE | CR |
|---|---|---|---|---|---|
| Attitude Towards Responsible Acquisition of Computer Behaviour | ATRACB1 | 0.993 | 0.001 | 0.798 | 0.938 |
| | ATRACB2 | 0.580 | 0.001 | | |
| | ATRACB3 | 0.938 | 0.001 | | |
| | ATRACB4 | 0.995 | 0.001 | | |
| Subjective Norms | SN1 | 0.998 | 0.001 | 0.833 | 0.936 |
| | SN2 | 0.677 | 0.001 | | |
| | SN3 | 0.949 | 0.001 | | |
| Perceived Behavioural Control | PBC1 | 0.964 | 0.001 | 0.655 | 0.897 |
| | PBC2 | 0.473 | 0.001 | | |
| | PBC3 | 0.961 | 0.001 | | |
| | PBC4 | 0.486 | 0.001 | | |
| | PBC5 | 0.983 | 0.001 | | |
| Responsible Acquisition of Behavioural Intention | RACBI1 | 0.825 | 0.001 | 0.605 | 0.821 |
| | RACBI2 | 0.758 | 0.001 | | |
| | RACBI3 | 0.749 | 0.001 | | |
| Responsible Acquisition of Computer Behaviour | RACB1 | 0.772 | 0.001 | 0.550 | 0.784 |
| | RACB2 | 0.810 | 0.001 | | |
| | RACB3 | 0.632 | 0.001 | | |
| Habits | HA1 | 0.993 | 0.001 | 0.849 | 0.943 |
| | HA2 | 0.759 | 0.001 | | |
| | HA3 | 0.993 | 0.001 | | |
| Egoistic | Ego1 | 0.952 | 0.001 | 0.796 | 0.951 |
| | Ego2 | 0.646 | 0.001 | | |
| | Ego3 | 0.647 | 0.001 | | |
| | Ego4 | 0.957 | 0.001 | | |
| | Ego5 | 0.653 | 0.001 | | |
| Altruistic | Altru1 | 0.665 | 0.001 | 0.826 | 0.948 |
| | Altru2 | 0.997 | 0.001 | | |
| | Altru3 | 0.673 | 0.001 | | |
| | Altru4 | 0.959 | 0.001 | | |
| Biospheric | Bio1 | 0.974 | 0.001 | 0.947 | 0.986 |
| | Bio2 | 0.974 | 0.001 | | |
| | Bio3 | 0.969 | 0.001 | | |
| | Bio4 | 0.976 | 0.001 | | |
| Environmental Concern | EC1 | 0.981 | 0.001 | 0.922 | 0.983 |
| | EC2 | 0.918 | 0.001 | | |
| | EC3 | 0.922 | 0.001 | | |
| | EC4 | 0.985 | 0.001 | | |
| | EC5 | 0.992 | 0.001 | | |
| Awareness of Consequences-Egoistic | AC1 | 0.992 | 0.001 | 0.981 | 0.994 |
| | AC2 | 0.997 | 0.001 | | |
| | AC3 | 0.982 | 0.001 | | |
| Awareness of Consequences-Altruistic | AC4 | 0.971 | 0.001 | 0.833 | 0.937 |
| | AC5 | 0.986 | 0.001 | | |
| | AC6 | 0.664 | 0.001 | | |

*(Continued)*

**Table 2.** (Continued)

| Construct | Items | Loadings | Sig. | AVE | CR |
|---|---|---|---|---|---|
| Awareness of Consequences- Biospheric | AC7 | 0.979 | 0.001 | 0.943 | 0.987 |
|  | AC8 | 0.993 | 0.001 |  |  |
|  | AC9 | 0.940 | 0.001 |  |  |
| Ascription of Responsibility | AR1 | 0.998 | 0.001 | 0.814 | 0.927 |
|  | AR2 | 0.687 | 0.001 |  |  |
|  | AR3 | 0.987 | 0.001 |  |  |
| Personal Norms | PN1 | 0.972 | 0.001 | 0.863 | 0.949 |
|  | PN2 | 0.609 | 0.001 |  |  |
|  | PN3 | 0.995 | 0.001 |  |  |

# 5. Discussion and implications

## 5.1 The extent of RACB in Malaysia

The study's findings show that Malaysians' RACB is still low, as indicated by the mean rating of 3.11. This means that most Malaysian consumers are opting to purchase non-green computers over green computers and are buying new computers although their present computers are still functional. This suggests that green-purchasing behaviour in Malaysia is still in its infancy compared with Western countries [71]. The reasons why Malaysian consumers opt to purchase computers responsibly are discussed as follows.

## 5.2 Extended TPB with Bio and Habits

The findings reveal that the root cause of RACB is the Bio value. The Bio value triggers the self-interest factor of Subjective Norms (SN) (H6), which triggers Habits (HA) (H4). Habits

**Table 3. Discriminant validity.**

|  | M | SD | AT RACB | SN | PBC | RACBI | EGO | AL TRU | BIO | EC | AC ego | AC bio | AC alt | AR | PN | HA | RACB |
|---|---|---|---|---|---|---|---|---|---|---|---|---|---|---|---|---|---|
| **AVE** |  |  | 0.953 | 0.948 | 0.939 | 0.605 | 0.939 | 0.956 | 0.942 | 0.990 | 0.981 | 0.958 | 0.943 | 0.985 | 0.966 | 0.849 | 0.562 |
| **ATRACB** | 4.050 | 0.800 | 0.976 |  |  |  |  |  |  |  |  |  |  |  |  |  |  |
| **SN** | 3.421 | 0.931 | 0.201 | 0.974 |  |  |  |  |  |  |  |  |  |  |  |  |  |
| **PBC** | 3.616 | 0.987 | 0.248 | 0.139 | 0.969 |  |  |  |  |  |  |  |  |  |  |  |  |
| **RACBI** | 3.085 | 0.477 | 0.050 | 0.138 | 0.067 | 0.778 |  |  |  |  |  |  |  |  |  |  |  |
| **EGO** | 2.983 | 1.210 | 0.01 | 0.074 | 0.084 | 0.057 | 0.969 |  |  |  |  |  |  |  |  |  |  |
| **ALTRU** | 3.624 | 0.897 | 0.425 | 0.401 | 0.159 | 0.160 | 0.035 | 0.978 |  |  |  |  |  |  |  |  |  |
| **BIO** | 2.814 | 1.260 | 0.045 | 0.184 | 0.051 | 0.149 | 0.349 | 0.093 | 0.971 |  |  |  |  |  |  |  |  |
| **EC** | 2.919 | 1.361 | 0.008 | 0.206 | 0.060 | 0.175 | 0.190 | 0.107 | 0.695 | 0.994 |  |  |  |  |  |  |  |
| **ACego** | 3.608 | 1.095 | 0.349 | 0.072 | 0.121 | 0.018 | 0.112 | 0.355 | 0.079 | 0.060 | 0.990 |  |  |  |  |  |  |
| **Acaltru** | 3.254 | 1.022 | 0.256 | 0.277 | 0.117 | 0.164 | 0.029 | 0.359 | 0.148 | 0.146 | 0.212 | 0.979 |  |  |  |  |  |
| **Acbio** | 3.630 | 1.028 | 0.368 | 0.050 | 0.119 | 0.093 | 0.152 | 0.309 | 0.061 | 0.183 | 0.464 | 0.184 | 0.971 |  |  |  |  |
| **AR** | 3.757 | 0.875 | 0.488 | 0.200 | 0.259 | 0.130 | 0.025 | 0.399 | 0.001 | 0.013 | 0.371 | 0.315 | 0.388 | 0.992 |  |  |  |
| **PN** | 3.620 | 0.826 | 0.479 | 0.259 | 0.168 | 0.121 | 0.004 | 0.487 | 0.049 | 0.060 | 0.332 | 0.407 | 0.302 | 0.470 | 0.983 |  |  |
| **HA** | 3.380 | 0.884 | 0.376 | 0.428 | 0.157 | 0.243 | 0.056 | 0.561 | 0.174 | 0.244 | 0.299 | 0.510 | 0.230 | 0.396 | 0.462 | 0.921 |  |
| **RACB** | 3.114 | 0.934 | 0.131 | 0.334 | 0.096 | 0.477 | 0.086 | 0.336 | 0.145 | 0.169 | 0.042 | 0.390 | 0.031 | 0.213 | 0.284 | 0.469 | 0.749 |

Note:

√AVE is shown in the diagonal values

Correlations for each construct are shown in the lower half of the table

M- Mean; SD–Standard Deviation

**Table 4. Standardised parameter estimates of the structural equation modelling.**

| Hypothesis | Path | | | Path coefficient | Standard Error | C.R. | Sig. Level | Supported |
|---|---|---|---|---|---|---|---|---|
| H1 | ATRACB | → | RACBI | -0.066 | 0.037 | 1.78 | P>0.05 | Not supported<br>The path is deleted as the relationship is insignificant, P>0.05. |
| H2a | SN | → | RACBI | 0.047 | 0.140 | 0.34 | P>0.001 | Not supported<br>The path is deleted as the relationship is insignificant, P>0.05. |
| H2b | SN | → | RACB | 0.505 | 0.035 | 14.43 | P<0.001 | Supported |
| H2c | SN | → | PN | 0.383 | 0.038 | 10.08 | P<0.001 | Supported |
| H2d | SN | → | AR | 0.357 | 0.038 | 9.39 | P<0.001 | Supported |
| H2e | SN | → | HA | 0.814 | 0.031 | 26.25 | P<0.001 | Supported |
| H3a | PBC | → | RACBI | 0.037 | 0.033 | 1.12 | P>0.05 | Not supported<br>The path is deleted as the relationship is insignificant, P>0.05. |
| H3b | PBC | → | RACB | 0.020 | 0.031 | 0.65 | P>0.05 | Not supported<br>The path is deleted as the relationship is insignificant, P>0.05. |
| H4 | RACBI | → | RACB | 0.510 | 0.031 | 16.45 | P<0.001 | Supported |
| H5a | EGO | → | EC | -0.059 | 0.026 | -2.26 | P<0.05 | Not Supported The path is deleted during model-trimming process due to beta value less than 0.10 |
| H5b | ALTRU | → | EC | 0.038 | 0.024 | 1.58 | P>0.05 | Not Supported<br>The path is deleted as the relationship is insignificant, P>0.05. |
| H5c | BIO | → | EC | 0.693 | 0.017 | 40.75 | P<0.001 | Supported |
| H6 | BIO | → | SN | 0.211 | 0.038 | 5.55 | P<0.001 | Supported |
| H7a | EC | → | ACego | -0.066 | 0.034 | -1.94 | P>0.05 | Not Supported<br>The path is deleted as the relationship is insignificant, P>0.05. |
| H7b | EC | → | ACaltru | 0.131 | 0.034 | 3.85 | P<0.001 | Not Supported<br>The path is deleted because a negative variance of the indicator exists. |
| H7c | EC | → | ACbio | 0.176 | 0.033 | 5.33 | P<0.001 | Supported |
| H8a | ACego | → | AR | 0.212 | 0.040 | 5.3 | P<0.001 | Not Supported.<br>The path became insignificant after non-significant path (refer to section 4.3) |
| H8b | ACaltru | → | AR | 0.204 | 0.032 | 6.38 | P<0.001 | Not Supported.<br>Negative variance exists. |
| H8c | ACbio | → | AR | 0.344 | 0.034 | 10.12 | P<0.001 | Supported |
| H9 | AR | → | PN | 0.319 | 0.035 | 9.11 | P<0.001 | Supported |
| H10 | PN | → | RACB | 0.014 | 0.039 | 0.35 | P>0.05 | Not Supported<br>The path is deleted because the relationship is insignificant, P>0.05. |
| H11 | HA | → | RACBI | 0.252 | 0.020 | 5.04 | P<0.001 | Supported |

subsequently triggers the Responsible Acquisition of Computers Behavioural Intention (RACBI) (H11), which finally triggers RACB. As a result, the original TPB is extended to explain the process of Malaysians' adoption of RACB.

The extended TPB shows that Malaysians' biospheric values affect their social acceptance of practising RACB. This is in line with the findings of Soyez [66]. The present study finds that the weak social support for practising RACB (Mean Rating [MR] = 3.42) is a result of the poor cultivation of biospheric values (MR = 2.81) (e.g., love of nature) among Malaysian consumers. This is evidenced in the findings of Bertsch et al. (2008). Malaysians do not really care about their environment (i.e., they possess low biospheric values) as evidenced by the low mean rating of the statement "protecting the natural environment is more important than creating economic growth and employment" (MR = 2.98) from the questionnaire.

The extended TPB illustrates that referents' opinions affect an individual's habit of practising RACB (path coefficient = 0.81; t = 6.50). For example, American-born software engineer, Rikin Gandhi, an employee of Microsoft Research in India, discovered that short, eight to ten-

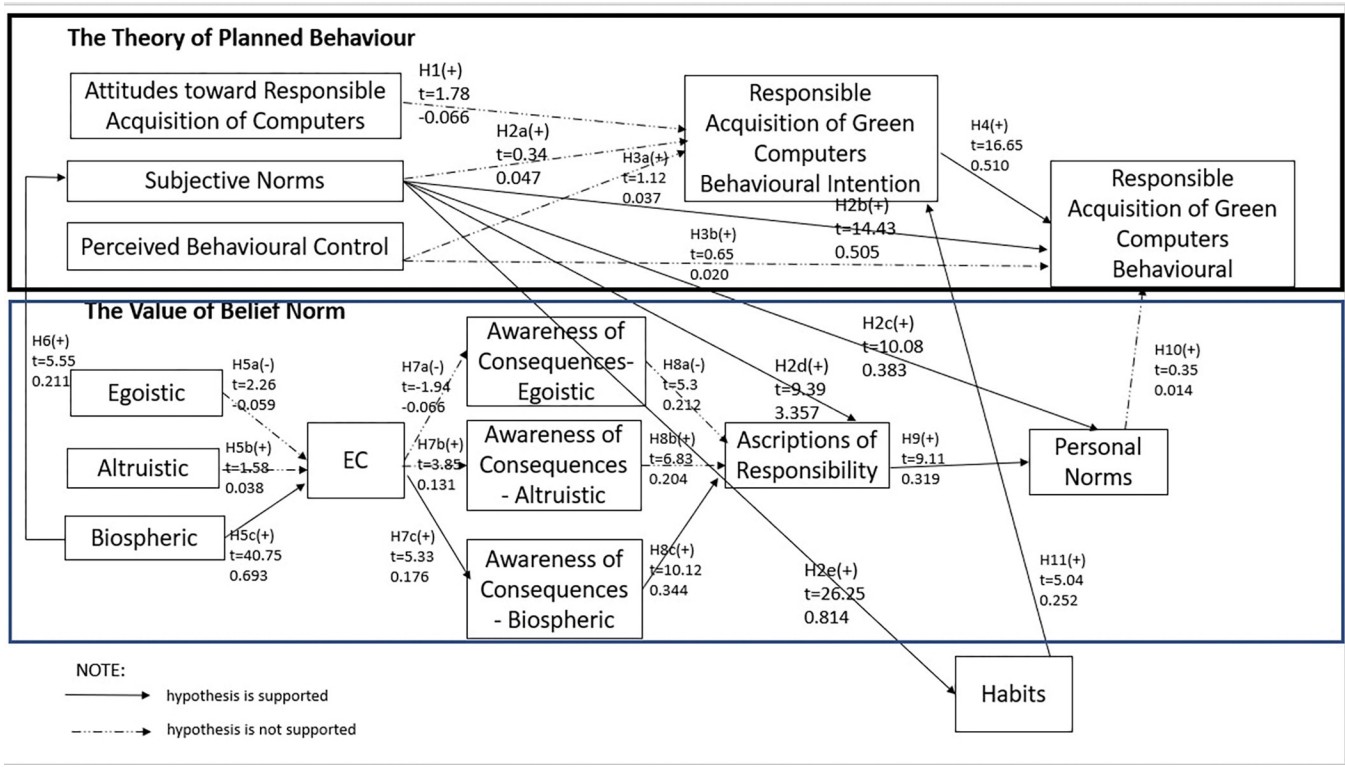

**Fig 2. Structural model for the final revised model.**

minute videos featuring local farmers talking about their experiences influenced their counterparts to adopt new agricultural habits because of a sense of belonging to the same group [51]. However, the habit of practising RACB among Malaysians is not strong (MR = 3.38) due to the weak influence of society. This concurs with Ahamad and Arrifin [33] who found that Malaysian consumers do not have a habit of practising sustainable consumption.

The present findings reveal that habits do not have a direct effect on RACB but indirectly affect it through the intentions (RACBI) of Malaysian consumers. The rationale is that the purchase of a computer is a high-cost behaviour that is performed irregularly (maybe once every one to two years), and in the unstable context of a changing environment where sales offers change and new green-computer features are introduced in the long interval between purchases. As a result, "intention" or "rational thinking" is necessary to guide individuals to practise RACB.

### 5.3 The extended value-belief-norm model and the subjective norms factor

The Bio value triggered the morality factor of Environmental Concern (EC)(H5c), which triggered Awareness of Consequences bio (ACbio)(H7c). ACbio subsequently triggered Ascription of Responsibility (AR) (H8c) and AR triggered Personal Norms (PN)(H9), forming a VBN causal chain. However, PN failed to explain RACB (H10). When the self-interest factor of SN was added to the VBN model, subjective norms were found to affect the morality factors of AR (H3) and PN (H2).

**5.3.1 Ego, Altru, and Bio affect Environmental Concern (EC) (H5a, H5b and H5c, respectively).** The confirmatory factor analysis substantiates that egoistic, altruistic, and biospheric values are distinct values, which is consistent with the findings of the prior studies

of Steg et al. [68], Garling et al.[69], and Alibeli and White [74]. This study finds that only biospheric values affect EC, whereas egoistic and altruistic values do not. Steg et al. [68] and De Groot and Steg [91] rationalised that individuals with egoistic values are concerned about the environment because of self-interest, while those with altruistic values are concerned with the welfare of others. As such, individuals with egoistic and altruistic values will not pay attention to the environment of the valued object (the biosphere), whereas individuals with biospheric values (who emphasise natural living, the biosphere, and ecosystems) will be affected by environmental concerns. The past research of De Groot and Steg [91] also substantiated that biospheric values become an important determinant of environmental concern in the context of pro-environmental behaviour. However, Malaysians do not have strong biospheric values (MR = 2.91), resulting in a low level of environmental concern (MR = 2.81). This is similar to the findings of Zwiers (2010), where Malaysian consumers are said to have low biospheric values and pay less attention to the environment as a result.

**5.3.2 EC affects ACego, ACaltru and ACbio (H7a, H7b and H7c, respectively).** The findings reveal that individuals' environmental concern from not practising RACB will affect their awareness of consequences to the biosphere, but not to others or themselves. This is because individuals will select congruent information and disregard or deny incongruent information. In other words, people who are concerned about the environment will pay more attention to the adverse consequences of environmental problems to the biosphere (such as $CO_2$ emissions that endanger plants, animals, and nature) and will disregard incongruent information that not performing RACB could also have harmful effects on society, the country, the next generation or themselves due to problems of toxic substances in the air and water and depletion of resources (electricity) from the purchase of non-green computers. The findings reveal that the respondents were not concerned about the environment (EC, MR = 2.81), which contributed to their vague beliefs about the impact that their computer-purchase decisions could have on the environment. Shahnaei [36] reported that Asians, including Malaysians, have a low level of environmental concern compared to people in Western countries, and their perception of the seriousness of environmental consequences is lower than that of Westerners. Westerners are concerned about the environment and view "green products" as a means of reducing pollution problems. Asians, however, view green products as costly specialty products and are not aware of the positive environmental consequences of buying them.

**5.3.3 ACego, ACaltru and ACbio affect AR (H8a, H8b and H8c, respectively).** Although Garling et al. [69] and Steg et al. [68] found that AC induces AR, these studies measured AC as a unidimensional instead of a multi-dimensional construct. As suggested by Lind et al. (2015), the present study has classified AC into ACego, ACaltru, and ACbio based on the object (the self, others, or the biosphere) that is valued. ACbio is found to be the predictor of AR while ACego and ACaltru do not affect it. This is because, from the findings of the present study, the consequences to the valued object are only to the biosphere, not to oneself or others. The present study reveals that Malaysians' awareness of the environmental consequences of not purchasing green computers is moderate (MR = 3.63), probably due to their lack of knowledge that they can reduce $CO_2$ emissions by purchasing green computers [34]. As a result of this uncertainty, Malaysian respondents do not have strong feelings about their responsibility for the negative environmental effects resulting from not practising RACB (MR = 3.75).

**5.3.4 AR affects PN (H9).** The present findings corroborate the VBN causal chain and the prior findings of Wall et al. [92] and Garling et al. [69] that personal norms are activated through AR. The present findings denote that Malaysians do not feel strongly that they are responsible for the negative environmental consequences of not acquiring green computers (MR = 3.65), thus leading to mediocre PN (MR = 3.62). This strengthens the findings of earlier research regarding Malaysians' lack of awareness of the consequences of their computer

purchase behaviour on the environment; they do not feel responsible for the consequences and this affects their moral obligation to practise RACB.

**5.3.5 Personal norms (PN) do not affect RACB (H10).** Various studies have substantiated that personal norms affect behavioural intention or behaviour directly in the context of pro-environmental behaviour. The findings of those earlier studies, however, contradict the present finding that personal norms (PN) do not exert any direct effects on behaviour. According to Abrahamse and Steg [93], a certain amount of planning, deliberation, and rational decision-making (behavioural intention) is required to induce behaviour, particularly high-cost behaviour where self-interest considerations will quickly displace morality factors. The displacement is extended to the RACB, a high-cost pro-environmental behaviour in the Malaysian context, and can be justified by the fact that Malaysian consumers are unwilling to pay more for green computers since such a practice is not a social norm. They will prioritise self-interest over morality factors and follow social norms by choosing non-green computers, disregarding the fact that such computers have harmful effects on the environment.

**5.3.6 Social norms (SN) affect RACB, AR, and PN (H1, H2, H3, respectively).** The present findings reveal that SN affect RACB directly, which is consistent with the findings of Wahid et al. [94]. This implies that RACB will be practised if it is widely accepted by Malaysian society, since what others (e.g., friends, family members, etc.) think is important in determining individual behaviour. This finding is consistent with the findings of Lee [95] and Abdullah et al. [46], who reasoned that an individual will behave according to what other people would morally approve or disapprove of, particularly in a culture of collectivism where norms come from close and concrete sources such as authority figures or abstract others without much thought. In other words, Malaysians practise RACB directly due to a collectivist culture that values obedience and social reciprocity and that motivates them to comply with the social expectations of close friends, family, or authority figures. The weak social support for practising RACB (MR for SN = 3.42) might be due to the "green purchase" concept still being new in Malaysia. This is supported by Nor Azila Mohd et al.'s [96] finding that only 30 per cent of study respondents were categorised as green product buyers with experience in purchasing such products. The effect of SN on AR is significant (path coefficient = 0.36; t = 5.25). In line with Mukama [53], although Malaysians rely on SN to determine their AR, weak social support (SN, MR = 3.42) has resulted in Malaysians' ascription of responsibility for the consequences of not practising RACB to be mediocre (MR = 3.75).

The current research reveals that social norms (SN) influence both actual behaviour and PN (path coefficient = 0.38; t = 6.35), concurring with the findings of Ahn et al. [97], Bamberg and Moser [30] and Valle et al. [72]. An individual's perception of whether a behaviour is right or wrong at the personal level (PN) is determined by his/her social group's expectations, so Malaysian consumers will learn if RACB is morally right or wrong (i.e., a personal norm) from the beliefs of their friends, family, and society (i.e., a social norm). The mediocre finding for SN (MR = 3.42) and RACB implies, however, that RACB is not strongly embraced by Malaysian society, which affects Malaysians' perceptions regarding their personal obligation of engaging in RACB. The result is that Malaysians do not have strong feelings that practising RACB is their responsibility.

## 5.4 Other non-significant TPB variables

The attitude towards the responsible acquisition of computers behaviour (ATRACB) does not influence RACBI. Although this is not aligned with the TPB model, some studies (e.g., [98]) found no relationship between attitude and pro-environmental behaviour. The present study finds that individuals in collectivist cultures are more pressured by SN than by attitude in their decision-making, confirming the findings of Soyez [66] and Khan et al. [29].

Additionally, social norms (SN) do not affect RACBI, which is inconsistent with the TPB model. Notably, social norms (SN) relate to individuals' perceptions of social pressure from people who are important to them. This influences individuals to behave (or not) in a certain manner and provides their motivation to comply with the views of such people [45]. The present findings reaffirm that SN could have a direct impact on actual behaviour, without going through intention, as a majority of Malaysian consumers have strong beliefs in social norms [63].

Finally, the effect of PBC on RACBI and RACB is insignificant. Ajzen [38] noted that the relative importance of PBC in the prediction of intention and behaviour varies across situations. PBC does not have any effect on intention (i.e., RACBI) due to Malaysian consumers' intentions being guided more by their habits, as reflected in the present findings. To reduce inconsistency pressures, Malaysians will observe if their current purchase intention is similar to their past behaviour (habits). In addition to this finding, this study reveals that PBC does not exert any effect on RACB. RACB could be a volitional control behaviour, wherein Malaysian consumers neither perceive that it is difficult nor easy for them to perform RACB in terms of the sufficiency of eco-labelling, pricing, etc as indicated by the MR of 2.89. Hence, PBC does not affect actual behaviour in the present context.

## 5.5 Theoretical implications

This study enriches the literature about responsible purchasing behaviour by addressing the absence of empirical studies examining the level of RCB and the antecedents of adopting RCB in the context of high-cost purchase decisions related to the acquisition of computers by individual consumers, particularly in emerging economy (i.e. Malaysia) comprised of low- and middle-income groups, fast-growing computer penetration and collectivist cultures [85].

This study provides holistic views relating to the integration of the complete TPB and VBN models with the habits variable in the high-cost pro-environmental behaviour domain, filling the literature gap where most of the scholars either dived into individual models like TPB [65, 43] or VBN [88] or extended individual models with additional variables [24, 99, 100]. It unveils the complex relationships among the two models and the habits variable. Thus, this contribution is unique.

The present findings confirm that a cognitive deliberative process occurs and that self-interest variables are significant when environmentally responsible behaviour is costly. Also, biospheric values are the main trigger of the TPB's self-interest variable and VBN's morality variable in forming decisions to adopt RACB. This further re-affirms that biospheric values are distinct from altruistic values in the context of pro-environmental behaviour [24, 43, 99]. SN are a direct predictor of RACB (TPB variable), PN (VBN variable), AR (VBN variable), and HA (TIB). This further substantiates that SN are a particularly important element in the context of developing countries with collectivist cultures. In addition, HA affects intention but not RACB directly. This implies that individuals' decisions will be affected by their HA in reasoning and assessing the costs and benefits of each alternative (i.e., RACBI) in the context of high-cost pro-environmental behaviour.

## 5.6 Practical implications

One practical implication of the findings of this research is that governments and environmental NGOs can make biospheric values more salient and instil them through informational strategies in formal and informal education campaigns to influence individuals' pro-environmental beliefs, intentions, and behaviour [65, 101]. For formal education, core subjects/syllabuses offered in elementary and secondary schools and tertiary education institutions could

incorporate environmental issues related to purchasing decisions. For informal environmental education, seminars, exhibitions, radio shows, drawing or colouring competitions, workshops, or the mass and social media (e.g., television, newspapers, the internet) can be used to create public awareness of the importance of biospheric values [101, 102].

Manufacturers can assist in informing consumers that their purchasing decisions are in line with biospheric values through voluntary labelling. According to a 2008 report by the Organisation for Economic Co-operation and Development (OECD), multi-criteria labels that compare products with others in the same category in terms of environmental impact throughout their life cycle, as well as single-issue labels like the Energy Star label that specify the environmental issue(s) addressed by the product (e.g. energy efficiency), are most useful in communicating complex information about a particular product to enable customers to make informed purchasing decisions. Moreover, consumers perceive that purchasing eco-labelled products is an environmentally friendly behaviour compared to purchasing conventional products [24]. Riding on this, marketers should be informed of the environmental benefits of Energy Star-labelled computers and deliver such information to the end consumer to enhance their understanding of the importance of practising RACB to protect the environment and make them aware that such behaviour is congruent with biospheric values.

Apart from strengthening biospheric values, the extended TPB also indicates that the relationships between SN and HA will elicit RACBI. Additionally, SN have a direct impact on RACB and the two VBN variables of PN and AR. With this in mind, governments could build strong social support for RACB among consumers via policy-making and social marketing. This could affect consumers' habits, their perception of responsibility for their actions on the environment, their moral obligation in practising RACB and their behaviour. Governments could limit the selling of generic PCs and provide subsidies to consumers who purchase green computers. Nguyen [43] and Zhang et al. [99] suggested the placement of advertisements in the mass media as part of information campaigns to educate the public that pro-environmental behaviour is morally and socially acceptable. This would motivate others (e.g., friends, family, government, and people close to consumers) to support RACB and build strong social norms, which will lead to a social change in HA, PN, AR and PN.

Lin et al [103] suggested that governments, retailers, and organisations could leverage mobile applications to promote sustainable responsible consumption. Given the success of the World Wildlife Fund Malaysia (WWF-Malaysia) and Microsoft Malaysia "Earth Hour Malaysia" smartphone apps (Microsoft, 2012), governments could allocate funds to WWF-Malaysia or Microsoft Malaysia or other relevant organizations to create and distribute similar apps to promote and educate Malaysians about RACB. These apps could reinforce social messages about the social expectations of performing RACB to achieve collective benefits (i.e., that RACB can reduce harmful environmental impact to benefit the next generation and society), thus providing strong social support and affecting consumers' AR and PN beliefs, habits, and actual behaviour.

## 6.0 Conclusion, limitations, and future studies

This study finds that the TPB and the VBN theory can be integrated with the habits variable to explain RACB. Biospheric values trigger subjective norms, subsequently leading to the formation of habits, which lead to the development of intentions and RACB. Biospheric values also trigger environmental concern that then generates awareness of consequences, followed by an ascription of responsibility that leads to the development of personal norms. Although personal norms were not found to affect RACB, subjective norms affect the ascription of responsibility, personal norms, and RACB.

The limitations of the study include the following. First, the study's domain is high cost/involvement pro-environmental behaviour at the product acquisition stage. Generalising the findings to other consumption stages (e.g., use and disposal) or other products should be done with caution until these findings have been replicated in other RCB contexts. Second, this study used only a quantitative research approach owing to time and financial constraints. Although this approach provides for the examination of the interrelationships between each of the factors and the responsible acquisition of computers behaviour, a mix of methods is warranted for future studies. Finally, this study focused only on the main effects of the TPB and VBN variables. Future studies should investigate the impacts of demographic variables such as age, gender, or income (as moderating variables) on the main relationships. This can provide meaningful information for manufacturers, marketers, and policymakers to facilitate the promotion of high-cost green computers among different groups of consumers.

## Supporting information

**S1 Appendix.**
(DOCX)

**S1 Dataset.**
(DOCX)

## Author Contributions

**Conceptualization:** W. H. Loo.

**Investigation:** Yuen Yee Yen.

**Supervision:** Paul H. P. Yeow.

**Validation:** Yuen Yee Yen.

**Writing – original draft:** W. H. Loo.

**Writing – review & editing:** Paul H. P. Yeow, Yuen Yee Yen.

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
