## [Decision Letter · Decision Letter 0]

8 Aug 2022

PONE-D-22-19976Antecedents of responsible acquisition of computers behaviour: Integrating theory of planned behaviour with value belief norm theory and habit variablePLOS ONE

Dear Dr. Yuen Yee Yen

Thank you for submitting your manuscript to PLOS ONE. After careful consideration, we feel that it has merit but does not fully meet PLOS ONE’s publication criteria as it currently stands. Therefore, we invite you to submit a revised version of the manuscript that addresses the points raised during the review process.

We look forward to receiving your revised manuscript.

Kind regards,

Tai Ming Wut

Academic Editor

PLOS ONE

Journal Requirements:

2. Please provide additional details regarding ethical approval in the body of your manuscript. In the Methods section, please ensure that you have specified the name of the IRB/ethics committee that approved your study. 

3. PLOS ONE does not copy edit accepted manuscripts (https://journals.plos.org/plosone/s/criteria-for-publication#loc-5). To that effect, please ensure that your submission is free of typos and grammatical errors, including the title.

4. lease provide additional details regarding participant consent. In the ethics statement in the Methods and online submission information, please ensure that you have specified (1) whether consent was informed and (2) what type you obtained (for instance, written or verbal, and if verbal, how it was documented and witnessed). If your study included minors, state whether you obtained consent from parents or guardians. If the need for consent was waived by the ethics committee, please include this information.

Reviewers' comments:

Reviewer's Responses to Questions

**Comments to the Author**

1. Is the manuscript technically sound, and do the data support the conclusions?

Reviewer #1: Yes

Reviewer #2: Yes

2. Has the statistical analysis been performed appropriately and rigorously? 

Reviewer #1: Yes

Reviewer #2: Yes

3. Have the authors made all data underlying the findings in their manuscript fully available?

Reviewer #1: Yes

Reviewer #2: Yes

4. Is the manuscript presented in an intelligible fashion and written in standard English?

Reviewer #1: Yes

Reviewer #2: Yes

5. Review Comments to the Author

Reviewer #1: This study focuses on discovering the antecedents of Responsible Acquisition of Computer Behaviour (RACB) which is a form of RCB. The study used TPB, VBN and habits variable to explain RACB. I have the following comments:

• More updated literature is required in 2.3 Perceived Behavioural Control and 2.5 Values and New Ecological Paradigm 2.6 Awareness of Consequences and 2.9 Habits. Please read the following papers:

o Wut, T. M., & Ng, P. M. L. (2022). Perceived CSR motives, perceived CSR authenticity, and pro-environmental behavior intention: an internal stakeholder perspective. Social Responsibility Journal, (ahead-of-print).

o Ng, P. M. L., & Cheung, C. T. Y. (2022). Why do young people do things for the environment? The effect of perceived values on pro-environmental behaviour. Young Consumers, (ahead-of-print).

o Yuriev, A., Dahmen, M., Paillé, P., Boiral, O., & Guillaumie, L. (2020). Pro-environmental behaviors through the lens of the theory of planned behavior: A scoping review. Resources, Conservation and Recycling, 155, 104660.

• Is habit a theory? You mentioned in the abstract that “This research aimed to investigate the antecedents of responsible acquisition of computers behaviour (RACB) among Malaysian consumers by using an integrated model of the three theories”. Please revise accordingly.

• The following hypotheses are missing but the arrows have been shown in Figure 1:

o Attitude � Responsible acquisition of green computers behavioural intention

o SN � Responsible acquisition of green computers behavioural intention

o PBC � Responsible acquisition of green computers behavioural intention

• Why 5-Likert scale is adopted? Any empirical support?

• Appropriate pre-test and pilot study

• Figure 1 and Figure 2 are difficult to follow. Please make a similar approach (especially H11, habit)

• For the implications, please divide them into (1) theoretical and (2) practical implications.

• Literature support in implications is insufficient. Please add more updated literature accordingly.

Reviewer #2: I appreciate that the authors have:

1. Chosen a hot topic that deserves timely research, discussion, and follow-up in line with the government’s initiatives and UN’s SDGs;

2. Demonstrated sufficient analysis and discussion of previous studies in the literature review;

3. Displayed innovation by integrating relevant theories to construct the conceptual framework;

4. Employed a large sample size (1,000) with data collected from 14 states in Malaysia, thereby enhancing power and representativeness;

5. Conducted thorough statistical analyses and explained results in detail; &

6. Displayed complex relationships among factors clearly in figures, which facilitated visualisation of the overall picture.

All in all, the study is of adequate standard for publication after minor revision. Please refer to following suggestions and questions for consideration.

Please consider:

1. Hiring a professional proof-reader (native speaker) to further enhance the readability of the article, e.g.,

a. Abstract: “The research question is how to encourage such behaviour”

b. Intro: “Their studies revealed that incorporating moral/personal norms into rational model (i.e. TPB) DO account”

c. Intro: “Moreover, most of the prior RCB combined TPB and VBN studies AT research mainly focused”

d. 2.6: “This study posits that the Malaysian consumer’s environmental concern will determineS”

e. “low-cost pro-environmental behaviour, its effect on high-cost behaviour remains unknown.”

->

Add “pro-environmental” after “high-cost”

f. 6. “and make them AWARE SUCH behaviour is congruent with Biospheric values.”

g. Please note that the above list is by no means exhaustive, please check for other typos/ errors

2. Intro and 2.1: “e.g. Ates (2020) – pro-environmental behaviour; Li et al. (2021) – purchase of organic food; Bamberg and Moser (2007) – pro-environmental behaviour; and Wall et al. (2007)’s studies – travel mode choices.”

->

Please elaborate a bit more instead of just stating the keywords e.g., “pro-environmental behaviour”

3. 2.1 – 2.4: “Therefore, this study does not need hypothesis testing on the relationship” and other parts with similar meaning e.g., “The relationship between SN and behavioural intention will not need to be tested in the current study as their relationship presented in the TPB model has already been validated across various contexts.”; “The aforementioned relationships as shown in the TPB model have empirical support, thus this study will not test their relationships.”; & “This study does not test the causal relationship between intention and actual behaviour since such relationships have been validated across different context.”

->

Despite the fact that previous studies have demonstrated positive findings, it may be advisable to test the hypotheses yourself once again.

4. 2.7: “…and the causal relationships between AC beliefs and AR beliefs were not empirically tested. Hence, the following hypotheses are tested in this study”

->

Can causality claims be made?

https://ftp.cs.ucla.edu/pub/stat_ser/r393-reprint.pdf

5. Were the questionnaire items in English? Did the respondents fully understand the questions?

6. Were only some of the items selected from the original questionnaires? Any implications on validity and reliability?

7. Please also include validity and reliability scores of the original questionnaires.

8. “As this study focused only on the main effects of the TPB and VBN variables, future studies could include demographic variables such as age, gender or income, which can be useful in providing meaningful information.”

->

Please clarify why such basic demographics were not collected.

6. PLOS authors have the option to publish the peer review history of their article (what does this mean?). If published, this will include your full peer review and any attached files.

Reviewer #1: **Yes: **Peggy Ng

Reviewer #2: No

---

## [Author Response · Author response to Decision Letter 0]

9 Nov 2022

Comment:

More updated literature is required in 2.3 Perceived Behavioural Control and 2.5 Values and New Ecological Paradigm 2.6 Awareness of Consequences and 2.9 Habits. Please read the following papers:

o Wut, T. M., & Ng, P. M. L. (2022). Perceived CSR motives, perceived CSR authenticity, and pro-environmental behavior intention: an internal stakeholder perspective. Social Responsibility Journal, (ahead-of-print).

o Ng, P. M. L., & Cheung, C. T. Y. (2022). Why do young people do things for the environment? The effect of perceived values on pro-environmental behaviour. Young Consumers, 23, 539-554.

o Yuriev, A., Dahmen, M., Paillé, P., Boiral, O., & Guillaumie, L. (2020). Pro-environmental behaviors through the lens of the theory of planned behavior: A scoping review. Resources, Conservation and Recycling, 155, 104660.

Response: 

We have updated the literature for Section 2.3, 2.5, 2.6, and 2.9, and included the following 8 most recent journal articles as our references. 

1. Ates, H. (2020). Merging theory of planned behaviour and value identity personal norm model to explain pro-environmental behaviour. Sustainable Production and Consumption, 24, 169-180.

2. Ates, H. (2021). Understanding students’ and science educators’ eco-labeled food purchase behaviours: Extension of Theory of Planned Behaviour with Self-identity, Personal Norm, Willingness to pay, eco-label knowledge. Ecology of Food and Nutrition, 1,1-16.

3. Ng, P. M. L., & Cheung, C. T. Y. (2022). Why do young people do things for the environment? The effect of perceived values on pro-environmental behaviour. Young Consumers, 23, 539-554.

4. Nguyen, T. N., Lobo, A. and Greenland, S. (2016). Pro-environmetnal purchase behaviour: The role of consumers’ biospheric values. Journal of Retailing and Consumer Services. 33, 98-108. 

5. Russell, S. V., Young, C. W., Unsworth, K. L., Robinson, C. (2017). Bringing habits and emotions into food waste behaviour. Resource Conservative Recycle, 125, 107–114.

6. Taufique, K. M. R., & Vaithianathan, S. (2018). A fresh look at understanding green consumer behaviour among young urban Indian consumers through the lens of theory of planned behaviour. Journal of Cleaner Production, 183, 46-55.

7. Wut, T. M., & Ng, P. M. L. (2022). Perceived CSR motives, perceived CSR authenticity, and pro-environmental behavior intention: an internal stakeholder perspective. Social Responsibility Journal, (ahead-of-print).

8. Yuriev, A., Dahmen, M., Paillé, P., Boiral, O., & Guillaumie, L. (2020). Pro-environmental behaviors through the lens of the theory of planned behavior: A scoping review. Resources, Conservation and Recycling, 155, 104660.

Comment:

• Is habit a theory? You mentioned in the abstract that “This research aimed to investigate the antecedents of responsible acquisition of computers behaviour (RACB) among Malaysian consumers by using an integrated model of the three theories”. Please revise accordingly.

Response: 

The sentences in the abstract has been revised to “This research aimed to investigate the antecedents of responsible acquisition of computers behaviour (RACB) among Malaysian consumers integrating the TPB, VBN, and Habits variable.”

Comment:

• The following hypotheses are missing but the arrows have been shown in Figure 1:

o Attitude � Responsible acquisition of green computers behavioural intention

o SN � Responsible acquisition of green comptuters behavioural intention

o PBC � Responsible acquisition of green computers behavioural intention

Response: 

In response to the comments above, the following literature reviews and hypotheses are added to the revised manuscript:

 Previous research validated and substantiated that positive attitudes will lead to greater behavioural intention positively in the context of RCB, e.g. Harland et al. (1999) – pro-environmental behaviour such as using unbleached paper, energy-saving light bulbs, turning off the faucet; Follow & Jobber (2000) – environmentally purchasing behaviour; Chan & Lau (2008) – green-purchase behavior; Nguyen et al., (2016)- purchase behaviour of energy efficient household appliances; Joshi & Rahman (2017) – sustainable purchasing behaviour; Ates, (2021) – eco-labeled food purchase. Thus, it is expected that Malaysian consumers who have positive attitude towards RACB will be more likely to engage themselves in RACB. Henceforth, the following hypothesis is postulated.

H1: Attitude towards RACB positively affect an individual’s intention of engaging RACB. 

SN is found have direct and significant effects on behavioural intentions to perform responsible purchasing behaviour (e.g. Dean et al., 2012; Paul et al., 2015; Muhammad et al., 2019). The strength of SN is relied on “normative belief” and “motivation to comply”. “Normative belief” refers to whether others think the individual should or should not perform responsible purchasing behaviour, whereas “motivation to comply” refers to the individual’s motivation to comply with society (Paul et al., 2015). Similarly, this study demonstrates that the consumers’ referent groups such as family members, friends or teachers’ opinions and expectations will influence their intention of practising RACB. As such, the following hypothesis is formed.

H2a: Subjective Norms positively affect an individual’s intention of engaging RACB. 

In line with the TPB, the PBC predicts specific responsible purchasing behaviour directly and indirectly through intentions (Azjen,1999). Prior studies supported the findings where PBC influenced pro-environmental responsible purchasing behavioural intention and actual responsible purchasing behaviour, ranging from recycling to green purchasing of household appliances (McCarty & Shrum, 2001). Researchers like Ates (2020); Yuriev et al (2020) delineated that consumers with high control over the barriers or discomfort in pro-environmental behaviour will likely involve themselves in pro-environmental behaviour. In a similar vein, Malaysian consumers who perceive that there are conducive external conditions like the availability of eco-label, advertisements related to RACB, green computers, and so forth will spur their RACB either directly or indirectly via behavioural intention. Therefore, the following hypotheses were formed. 

H3a: Perceived behavioural control positively affects an individual’s intention of engaging RACB.

Comment:

• Why 5-Likert scale is adopted? Any empirical support?

Response:

The following explanation is added to the revised manuscript:

The items measuring the TPB and VBN variables were assessed on a five-point Likert scale, which have been validated in previous studies (Sawitri et al., 2015; Fu et al., 2021). Revilla et al. (2014)’s findings evidenced that a five-point scale is the best scaling technique for agree and disagree questions in terms of data quality and response rate. 

References that support the use of 5-Likert scale are also added to the revised manuscript:

1. Sawitri, D. R., Hadiyanto, H. and Hadi, S. P. (2015). Pro-environmental behaviour from a Social Cognitive Theory Perspective. Procedia Environmental Sciences. 23, 27-33.

2. Fu, W., Zhou, Y., Li, L. and Yang, R. (2021). Understanding household electricity-saving behaviour: Exploring the effects of perception and cognition factors. Sustainable Production and Consumption, 28, 116-128. 

3. Revilla, M. A., Saris, W. M. and Krosnick, J. A. (2014). Choosing the number of categories in Agree-Disagree scales. Sociological Methods & Research, 43, 73-97.

Comment

• Figure 1 and Figure 2 are difficult to follow. Please make a similar approach (especially H11, habit)

Response: 

Both figures have been re-drawn. Figure 1 reflects the combination of TPB, and VBN models together with one additional variable (habits). We’ve included the hypotheses Attitude->RACBI, SN-> RACBI, PBC->RACBI and RACBI->RACB accordingly. Changes have been made for Figure 2 as well.

Comment

• For the implications, please divide them into (1) theoretical and (2) practical implications.

Response: 

The following theoretical and practical implications have been discussed in section 5.4 and 5.5.

5.4 Theoretical Implications

The literature in responsible purchasing behaviour studies is enriched since there is an absence of an empirical study that examines the level of RCB and the antecedents of adopting RCB with respect to computer acquisition (high-cost purchase decision) by individual consumers, particularly in emerging economies, with fast-growing computer penetration (Worldometers, n.d) in a collectivist culture comprise low-income and middle-income groups. 

 The analysis of this study provides holistic views relating to integrating the TPB, and VBN models with habit variables in the high-cost pro-environmental behaviour domain. Thus, filling the literature gaps where most of the scholars either dived into individual models like TPB (e.g., Lago, 2020; Kang & Moreno, 2020) or VBN (e.g. Sharma and Gupta, 2020), or extended individual models with additional variables (e.g. Ates, 2021). The present findings validated that the cognitive deliberative process occurs, and self-interest variables are significant wherein the environmental behaviour is costly. Also, biospheric value is the main factor that triggered the TPB (self-interest aspect) and VBN (morality aspect) variables in forming decisions to adopting RACB. This further re-affirmed the biospheric value is distinct from altruistic value in the context of pro-environmental behaviour (Nguyen, 2016; Ates, 2021). SN is direct predictor of RACB (TPB variable), PN (VBN variable), AR (VBN variable) and HA (TIB). This further substantiated that SN is a very important element in the context of developing countries with collectivist cultures. Besides, HA are found to affect intention but not RACB directly. This implies that in high-cost pro-environmental behaviour, individuals’ decisions will be affected by their HA through their reasoning and assessing of the costs and benefits of each alternative (i.e. RACBI).

 5.5 Practical Implications

Governments and environmental NGOs can make biospheric values more salient and instill them through informational strategies in formal and informal education, which will ultimately influence individuals’ pro-environmental beliefs, intentions and behaviour (Nguyen et al., 2019; Dharmesti et al., 2020). For formal education, core subjects/syllabuses offered in elementary, secondary and tertiary education could incorporate environmental issues related to purchase decisions. For informal environmental education, seminars, exhibitions, radio shows, drawing or colouring competitions, workshops or the mass media such as television, newspapers, the Internet or during social interaction could be used to create public awareness about the importance of biospheric values (Hassan & Pudin, 2011; Dharmesti et al., 2020). 

Manufacturers can assist consumers to be informed that their purchasing decisions are in line with biospheric values through voluntary labelling. According to the Organisation for Economic Co-operation and Development (OECD) 2008 report, multi-criteria labels, which compare products with others in the same category in terms of environmental impact throughout their life cycles, as well as single-issue labels (e.g. the Energy Star label) which specifies environmental issue addressed by the product (e.g. energy efficiency), are most useful in communicating complex information about a particular product, thus, enabling customers to make informed choices. Moreover, consumers perceive that purchasing eco-label products is an environmentally friendly behaviour compare to conventional products (Ates, 2021). In this regards, marketers should be informed of the environmental benefits contributed by Energy Star-label computers. Hence, such information could be delivered to the final consumers to enhance their understanding of the importance of practising RACB to protect the environment, and make them aware such behaviour is congruent with biospheric values. 

Apart from strengthening the biospheric values, the extended TPB also indicates the relationships between SN and HA will elicit RACBI. Additionally, the SN is found to have direct impact on RACB and the two VBN variables, i.e. PN and AR. The government could build a strong social support for RACB among consumers, which could affect others’ habits, perceptions of their responsibility for their actions on the environment, their personal moral obligation in practising RACB and their behaviour, via policy-making and social marketing. Government could limit the selling of cloned PCs and provide subsidies to those who purchase green computers. Nguyen (2016) and Zhang et al. (2017) suggested advertisements in the mass media, and information campaigns or education could educate the public to ensure that pro-environmental behaviour is morally and socially acceptable. This will motivate others (e.g. friends, family, government and people around the individuals) to support RACB. Subsequently, building strong social norms, which will lead to a social change of HA, PN, AR and PN.

Lin et al (2022) suggested that governments, retailers or organisations could leverage mobile applications to promote sustainable responsible consumption. Likewise, Wide Fund for Nature (WWF)--Malaysia and Microsoft Malaysia smartphone application (app) on “Earth Hour Malaysia” was a success (Microsoft, 2012). Perhaps, the government could allocate funds to WWF-Malaysia or Microsoft Malaysia or other relevant organizations to duplicate similar apps to promote and educate Malaysians about RACB. These apps could reinforce social messages with regard to the social expectations on performing RACB for the collectivist benefits, such as RACB could reduce harmful environmental impact to the next generation and society, thus providing strong social support and in turn affecting consumers’ AR and PN beliefs, habits and actual behaviour. 

Comment

• Literature support in implications is insufficient. Please add more updated literature accordingly.

Response: We have updated the literature by adding the recent references as listed below.

1. Ates, H. (2021). Understanding students’ and science educators’ eco-labeled food purchase behaviours: Extension of Theory of Planned Behaviour with Self-identity, Personal Norm, Willingness to pay, eco-label knowledge. Ecology of Food and Nutrition, 1, 1-16.

2. Dharmesti, M., Merrilees, B., & Winata, L. (2020). “I’m mindfully green”: Examining the determinants of guest pro-environmental behaviors (PEB) in hotels. Journal of Hospitality Marketing & Management, 1, 1–18. 

3. Lin, Y.T., Tseng, T. H., Chang, A., and Yang, C.C. (2022). A value adoption approach to sustainable consumption in retail stores. International Journal of Retail & Distribution Management, 50, 1412-1434.

4. Nguyen, T. N., Lobo, A. and Greenland, S. (2016). Pro-environmetnal purchase behaviour: The role of consumers’ biospheric values. Journal of Retailing and Consumer Services. 33, 98-108. 

5. Zhang, X.J., Geng, G.J. and Sun, P (2017). Determinants and implications of citizens’ environmental complaint in China: Integrating theory of planned behaviour and norm activation model. Journal of Cleaner Production, 166, 148-156

Comment:

1. Hiring a professional proof-reader (native speaker) to further enhance the readability of the article, e.g.,

a. Abstract: “The research question is how to encourage such behaviour”

b. Intro: “Their studies revealed that incorporating moral/personal norms into rational model (i.e. TPB) DO account”

c. Intro: “Moreover, most of the prior RCB combined TPB and VBN studies AT research mainly focused”

d. 2.6: “This study posits that the Malaysian consumer’s environmental concern will determine”

e. “low-cost pro-environmental behaviour, its effect on high-cost behaviour remains unknown.”

->

Add “pro-environmental” after “high-cost”

f. 6. “and make them AWARE SUCH behaviour is congruent with Biospheric values.”

g. Please note that the above list is by no means exhaustive, please check for other typos/ errors

Response: 

A professional proofreader is hired to proofread this manuscript to address these proofreading issues. Therefore, comments from number 1a to number 1g have been addressed accordingly. 

2. Intro and 2.1: “e.g. Ates (2020) – pro-environmental behaviour; Li et al. (2021) – purchase of organic food; Bamberg and Moser (2007) – pro-environmental behaviour; and Wall et al. (2007)’s studies – travel mode choices.”

->

Please elaborate a bit more instead of just stating the keywords e.g., “pro-environmental behaviour”

Response: 

The following explanations were added to the manuscript: 

Responsible consumption behaviour (RCB) is a pro-environmental behaviour and was defined as the “acquisition, consumption and disposition of goods, services, time and ideas by decision-making units without harming the environment or society” (Stancu, 2011). Thogersen (1999) highlighted that it is worth studying RCB that significantly affects environmental quality, particularly at the acquisition stage, as the right responsible purchasing decision could have the potential to reduce and eliminate environmental harm in the later stages of the consumption cycle. Joshi and Rahman (2015) supported this claim by highlighting that 40% of environmental harm is caused by irresponsible consumption through the purchase of non-sustainable products. 

RACB is defined as making a computer purchase without harming the environment such as buying Electronic Product Environmental Assessment Tools (EPEAT)-compliant computers. EPEAT is a standard certified by Green Electronics Council for computer product that contains less or no toxic content (e.g. heavy metals), consumes less energy, and is recyclable and easily upgradeable to extend its lifespan (Greenelectronicscouncil.org, 2018). 

Comment:

3. 2.1 – 2.4: “Therefore, this study does not need hypothesis testing on the relationship” and other parts with similar meaning e.g., “The relationship between SN and behavioural intention will not need to be tested in the current study as their relationship presented in the TPB model has already been validated across various contexts.”; “The aforementioned relationships as shown in the TPB model have empirical support, thus this study will not test their relationships.”; & “This study does not test the causal relationship between intention and actual behaviour since such relationships have been validated across different context.”

->

Despite the fact that previous studies have demonstrated positive findings, it may be advisable to test the hypotheses yourself once again.

Response: 

Relationships between TPB variables were included in the hypotheses (refer to H1 to H4 below) and tested in this study. 

H1: Attitude towards RACB positively affect an individual’s intention of engaging RACB. 

H2a: Subjective Norms positively affect an individual’s intention of engaging RACB. 

H2b: Subjective Norms positively affect an individual’s engagement in RACB.

H2c: Subjective Norms positively affect an individual’s Personal Norms in RACB. 

H2d: Subjective Norms positively affect an individual’s Ascription of Responsibility in RACB.

H2e: Subjective Norms positively affect an individual’s Habits in practising RACB.

H3a: Perceived behavioural control positively affects an individual’s intention of engaging RACB.

H3b: Perceived behavioural control positively affects an individual’s engagement in RACB. 

H4: Behavioural intention positively affects an individual’s engagement in RACB. 

Comment:

4. 2.7: “…and the causal relationships between AC beliefs and AR beliefs were not empirically tested. Hence, the following hypotheses are tested in this study”

->

Can causality claims be made?

https://ftp.cs.ucla.edu/pub/stat_ser/r393-reprint.pdf

Response: 

The following explanations were added to the manuscript: 

Stern et al. (1993) stated that awareness of adverse consequences (AC) will lead to AR. Individuals with high AC are presumed to become aware of the extensive and specific consequences of possible actions, and to adopt the perspective of those who will be affected when weighing decisions. Since AC beliefs are positively related to AR beliefs, i.e. individuals who are more aware of the adverse consequences of their action will feel more responsible to reduce the threats resulting from their action (Steg et al., 2005; Sahin, 2013), the following hypotheses are tested in this study:

H8a: An individual’s Awareness of Consequences to self positively affects his or her Ascription of Responsibility for acquiring computers responsibly.

H8b: An individual’s Awareness of Consequences to others positively affects his or her Ascription of Responsibility for acquiring computers responsibly.

H8c: An individual’s Awareness of Consequences to the biosphere positively affects his or her Ascription of Responsibility for acquiring computers responsibly.

Comment:

5. Were the questionnaire items in English? Did the respondents fully understand the questions?

Response: 

Yes. The questionnaire items are in English. The respondents can understand the question since Malaysia adopts a bilingual system of education. Notably, English language is a compulsory subject at primary, secondary and tertiary education levels (Darmi and Albion, 2013)

Darmi, R., & Albion, P. (2013). English language in the Malaysian education system: Its existence and implications. In Proceedings of the 3rd Malaysian Postgraduate Conference (MPC 2013) (pp. 175-183). Education Malaysia.

Comment:

6. Were only some of the items selected from the original questionnaires? Any implications on validity and reliability?

Response: 

Yes, some of the items were selected/adapted from the original questionnaires to suit this study. Appendix A shows the measure for the dependent variable, i.e. RACB, which consists of 3 items adapted from Ajzen (1991) and Murugesan (2008). Appendix B shows the measures for all TPB factors, which consist of 15 items adapted from Ajzen (1991), Boldero (1995) and Tonglet et al. (2004), and Murugesan (2008). Appendix C shows the measures for all VBN factors, which consist of 33 items adapted from Stern et al. (1999) and Steg et at. (2005). Appendix D shows the measures of habit factor, which consists of 3 items adapted from Venkatesh et al., (2012) and Triandis (1980). All items are measured using Likert 5-point scale, i.e. strongly disagree (1), disagree (2), neutral (3), agree (4) and strongly agree (5). The items were adopted from past studies. Some wording was changed to suit the present context. The validity and reliability of the constructs were tested using confirmatory factors analysis (CFA) and Cronbach’s alpha. In the final model, the composite reliability (CR) for all factors is above 0.70 which shows acceptable internal consistency (Nunnally & Bernstein, 1994). The Average Variance Extracted (AVE) for all factors is above 0.5 which shows acceptable convergent validity. Table 3 shows the √AVE values (see the diagonal values) of each construct is greater than the correlation values in the same row. This shows high discriminant validity. There were no multicollinearity issue in the model since all the correlations are below 0.8 (Sekaran, 2000). Cronbach Alpha (i.e. a reliability test statistic) was calculated for all factors and they were above the threshold value of 0.7 (Sekaran, 2000). Thus, there was no implication on the validity and reliability.

Comment:

7. Please also include validity and reliability scores of the original questionnaires.

Response:

The following explanation is added:

Prior to the pilot study, the original questionnaires were pretested for content validation by 10 experts who are researchers, lecturers and professor. Based on their feedback, amendments were made on the questionnaire content, format, terms, ease of completion and comprehensibility. A pilot study was then conducted with 100 questionnaires distributed in public places. Cronbach Alpha (i.e. a reliability test statistic) was calculated for all factors and they were above 0.7. 

Comment:

8. “As this study focused only on the main effects of the TPB and VBN variables, future studies could include demographic variables such as age, gender or income, which can be useful in providing meaningful information.”

Please clarify why such basic demographics were not collected.

Response: 

In fact, the basic demographic variables were collected in this study. However, this study focused only on the main effects of the TPB and VBN variables. Due to the limited space, we plan in the future studies to analyze the impacts of demographic variables such as age, gender or income, as moderating variables on main relationships, which may be useful in providing meaningful information for manufacturers, marketers and policy makers to facilitate them to promote high-cost green computers among consumers from different groups.

---

## [Decision Letter · Decision Letter 1]

17 Jan 2023

PONE-D-22-19976R1Antecedents of the responsible acquisition of computers behaviour: Integrating the theory of planned behaviour with the value-belief-norm theory and the habits variablePLOS ONE

Dear Yuen Yee Yen,

Thank you for submitting your manuscript to PLOS ONE. After careful consideration, we feel that it has merit but does not fully meet PLOS ONE’s publication criteria as it currently stands. Therefore, we invite you to submit a revised version of the manuscript that addresses the points raised during the review process.

We look forward to receiving your revised manuscript.

Kind regards,

Tai Ming Wut

Academic Editor

PLOS ONE

Reviewers' comments:

Reviewer's Responses to Questions

**Comments to the Author**

1. If the authors have adequately addressed your comments raised in a previous round of review and you feel that this manuscript is now acceptable for publication, you may indicate that here to bypass the “Comments to the Author” section, enter your conflict of interest statement in the “Confidential to Editor” section, and submit your "Accept" recommendation.

Reviewer #2: All comments have been addressed

Reviewer #3: (No Response)

2. Is the manuscript technically sound, and do the data support the conclusions?

Reviewer #2: Yes

Reviewer #3: Partly

3. Has the statistical analysis been performed appropriately and rigorously? 

Reviewer #2: Yes

Reviewer #3: No

4. Have the authors made all data underlying the findings in their manuscript fully available?

Reviewer #2: Yes

Reviewer #3: Yes

5. Is the manuscript presented in an intelligible fashion and written in standard English?

Reviewer #2: Yes

Reviewer #3: No

6. Review Comments to the Author

Reviewer #2: (No Response)

Reviewer #3: About model and originality of the paper: The authors select “RACB” as research subject on the topic of green computer purchase behavior in Malaysia. However, the authors don’t explain why these customers should be considered individually and don’t introduce any new indicators in this research filed. Also, the research model in the study is so complicated which not suitable for the SEM model. it may be necessary for the author to simplify the model or to consider using a different type of statistical analysis.

Literature review: The literature part is superficial and shallow even in the revised version. For example, the paper does not provide much detail or depth on the topic that what the weaknesses associated with the use of a single theory by the other authors one by one and the contributions to integrate TPB variables with VBN variables to explain RCB. Moreover, some updated and related literature about TPB & responsible (purchase) behavior were not mentioned/cited yet. I suggest the author(s) to consider reading and citing the following impactful, and updated literature; sure the authors can cite other similar papers not limit to listed ones.

• Gao, L., Bai, X. (2014). A unified perspective on the factors influencing consumer acceptance of internet of things technology. Asia Pacific Journal of Marketing and Logistics, 26(2), 211-231.

• Liu, M., Liu, Y., Mo, Z. (2020). Moral norm Is the key: An extension of the theory of planned behaviour (TPB) on Chinese consumers’ green purchase intentions, Asia Pacific Journal of Marketing and Logistics, 32(8), 1823-1841.

• Kumagai, K. (2021). Sustainable plastic clothing and brand luxury: a discussion of contradictory consumer behaviour, Asia Pacific Journal of Marketing and Logistics, 33(4), 994-1013

• Liu, M., Liu, Y., Mo, Z., Zhao, Z., Zhu, Z. (2020). How CSR influences customer behavioural loyalty in the Chinese hotel industry, Asia Pacific Journal of Marketing and Logistics, 32(1), 1-22.

• Wang, Y., Ko, E., & Wang, H. (2022). Augmented reality (AR) app use in the beauty product industry and consumer purchase intention. Asia Pacific Journal of Marketing and Logistics, 34(1), 110-131.

• Mo. Z., Liu M., Liu, Y. (2018). Effects of functional green advertising on self and others, Psychology & Marketing. 35(5), 368-382.

Methodology and result presentation: The process of data analysis in this paper is too simple and rough. The author needs to present each step of data analysis in the concise way. Also, the title of table 4 need to be revised. The author didn’t presents the direct and indirect effects clearly. Please enhance the mentioned parts.

The integration of the TPB and VBN models cannot be seen as the theoretical contribution. Without having enough implications/contributions associating with your core concepts (TPB or VBN), the value of the work will be much weakened. Please enhance your implications/contributions parts in resubmission.

There is a lot of inappropriate/non-typical English/nonsense expressions, the language is not academic-oriented enough. Some tables in the paper should be formatted consistently with three-line format. Before re-submission, I strongly suggest the author need to send the draft to proofreader/copy-editor for improvement.

7. PLOS authors have the option to publish the peer review history of their article (what does this mean?). If published, this will include your full peer review and any attached files.

Reviewer #2: No

Reviewer #3: No

---

## [Author Response · Author response to Decision Letter 1]

2 Mar 2023

Antecedents of the responsible acquisition of computers behaviour: Integrating the theory of planned behaviour with the value-belief-norm theory and the habits variable

Response to Comments

Reviewer #3: 

Comment

About model and originality of the paper: The authors select “RACB” as research subject on the topic of green computer purchase behavior in Malaysia. However, the authors don’t explain why these customers should be considered individually and don’t introduce any new indicators in this research filed. Also, the research model in the study is so complicated which not suitable for the SEM model. it may be necessary for the author to simplify the model or to consider using a different type of statistical analysis.

Response: 

Thank you for the comments. 

i. We included the following justifications for choosing individual consumers as the sample in Section 3.4. 

“Individual consumers were chosen as the unit of analysis because 40% of environmental degradation is attributed to individual consumers’ irresponsible consumption of non-sustainable products (Joshi and Rahman, 2017). ”

ii. Why new indicators/constructs are not introduced?

As it is, there are already sufficient 15 constructs from the Theory of Planned Behaviour, Value Belief Norms theory and habit construct. There is no need of adding new construct. There is no need of developing new variables as the existing theories (TPB and VBN) can explain RACB.

iii. Regarding the appropriateness of SEM technique, we added the below justifications as to why SEM technique is adopted in Section 3.6. Therefore, we didn’t change the model or use other statistical analysis technique. 

“SEM tests a large number of independent and dependent constructs statistically in a simultaneous analysis to develop a structural model, which is a plausible representation of the relations between all variables (Hair et al., 2014). SEM is a comprehensive approach of data analysis whereas traditional multivariate procedures are incapable of either assessing and correcting measurement error. SEM provides explicit estimates of the measurement error through confirmatory hypothesis-testing approach (Hair et al., 2014). Many researchers like Gao and Bai (2014); Kiatkawsin and Han (2017); Vincent et al. (2022); Nguyen (2022) etc. validated the causality among the independent and dependent constructs in the integrative research framework via SEM technique.”

• Hair, J. F., Jr., Black, W. C., Babin, B. J., & Anderson, R. E. (2014). Multivariate data analysis (7th ed.). Essex: Pearson Education.

• Gao, L., Bai, X. (2014). A unified perspective on the factors influencing consumer acceptance of internet of things technology. Asia Pacific Journal of Marketing and Logistics, 26(2), 211-231.

• Kiatkawsin, K., & Han, H. (2017). Young travelers' intention to behave pro-environmentally: Merging the value-belief-norm theory and the expectancy theory. Tourism Management, 59, 76-88.

• Vincent, K.E., Shuliang, Z., Benjamin, K. L (2022). Investigating household waste separaction behaviour: the salience of an integrated norm activation model and the theory of planned behaviour, Journal of Environment Planning and Management.

• Nguyen, T. P. L. (2022). Intention and behaviour toward bringing your own shopping bags in Vietnam: integrating theory of planned behaviour and norm activation model”, Journal of Social Marketing, 12(4), 395-419. 

iv. it may be necessary for the author to simplify the model

In fact, in the final revised model of SEM as shown in Fig. 2, the model is simplified from the original 15 constructs to 9 constructs as 6 constructs do not have any significant relationships., i.e. Attitude Towards Responsible Acquisition of Computers, Perceived Behavioural Control, Egoistic, Altruistic, Awareness of Consequences – Egoistic and Awareness of Consequences – Altruistic.

Comment:

Literature review: The literature part is superficial and shallow even in the revised version. For example, the paper does not provide much detail or depth on the topic that what the weaknesses associated with the use of a single theory by the other authors one by one and the contributions to integrate TPB variables with VBN variables to explain RCB. Moreover, some updated and related literature about TPB & responsible (purchase) behavior were not mentioned/cited yet. I suggest the author(s) to consider reading and citing the following impactful, and updated literature; sure the authors can cite other similar papers not limit to listed ones.

• Gao, L., Bai, X. (2014). A unified perspective on the factors influencing consumer acceptance of internet of things technology. Asia Pacific Journal of Marketing and Logistics, 26(2), 211-231.

• Liu, M., Liu, Y., Mo, Z. (2020). Moral norm Is the key: An extension of the theory of planned behaviour (TPB) on Chinese consumers’ green purchase intentions, Asia Pacific Journal of Marketing and Logistics, 32(8), 1823-1841.

• Kumagai, K. (2021). Sustainable plastic clothing and brand luxury: a discussion of contradictory consumer behaviour, Asia Pacific Journal of Marketing and Logistics, 33(4), 994-1013

• Liu, M., Liu, Y., Mo, Z., Zhao, Z., Zhu, Z. (2020). How CSR influences customer behavioural loyalty in the Chinese hotel industry, Asia Pacific Journal of Marketing and Logistics, 32(1), 1-22.

• Wang, Y., Ko, E., & Wang, H. (2022). Augmented reality (AR) app use in the beauty product industry and consumer purchase intention. Asia Pacific Journal of Marketing and Logistics, 34(1), 110-131.

• Mo. Z., Liu M., Liu, Y. (2018). Effects of functional green advertising on self and others, Psychology & Marketing. 35(5), 368-382.

Response: 

Thanks for the comments.

We have added the following new paragraphs to enhance the richness of the literature review regarding the TPB theory in terms of its applications, strengths and weakness along with the shortcoming of relying on single theory. We also have updated the literature review with the latest published articles, including the suggested articles. Please refer to the introduction (Section 1):

“Many studies have been conducted to discover the antecedents of RCB using various theories and factors such as the theory of planned behaviour (TPB), the value-belief-norm (VBN) theory, and the habitual factor (Gao& Bai, 2014; Liu, Liu & Mao, 2020; Kumagai, 2021; Mo, Liu & Liu, 2018). The TPB articulates how individuals make reasoned choices and choose alternatives with the greatest benefits against the lowest costs in serving their self-interest, wherein the behaviour is determined by behavioural intention that lies on individual positive assessment on the behaviour (attitude), social pressure adopting the behaviour (subjective norm) and perceived ease of engaging the behaviour (perceived behavioural control) (Kumagai, 2021). 

Prior studies substantiated that the TPB could serve as a basic model in explaining RCB encapsulating energy saving (Gao & Bai, 2014; Gao et al., 2017; Liu, Liu & Mao, 2020; Ru et al., 2018), recycling (Wan et al., 2017), purchasing organic food (Yazdanpanah and Fourouzani, 2015), green purchase (Kumagai, 2021; Mo, Liu & Liu, 2018), etc. However, the TPB owns its limitation i.e., TPB is a self-interest theory that primarily comprises rational predictors while omitting other relevant RCB contextual factors like moral obligation, habits, etc. (Yurie et al., 2020); thus, affecting the predictive power of the theory. This is evidenced in Yazdanpanah and Fourouzani (2015)’s findings, where the predictive power of the TPB became higher from 56% to 65% after adding moral norms and self-identity to the original TPB. Additionally, Gao et al. (2017)’s studies revealed that the explanatory power of TPB raised from 22.6% to 34.9%, and the significant predictors of individual energy-saving behaviour include all the TPB variables (except subjective norm) and the added variables i.e., descriptive norms and personal norms. Wan et al. (2017)’s studies found two additional variables, i.e., awareness of consequences and moral norms, increased the predictive power of the TPB in predicting recycling intention. Liu et al. (2019) revealed the extended TPB model performed well in green purchase intention setting. All these findings shed light for future RCB research that extending TPB is required as the theory fails to consider the moral aspect; thus, it cannot fully explain the behaviour. 

RCB is a pro-social behaviour which is not merely predicted by TPB variables, but also could be well-explained by VBN theory, wherein the VBN theory proposes that altruistic behaviour is triggered by the morality factors such as personal norms, ascriptions of responsibility, awareness of consequences, new ecological paradigm, and values. However, VBN theory does not cover the self-interest predictors that are captured by TPB variables. So far, most RCB researches focus mainly on extending TPB with one or a few variables instead of integrating the whole TPB with the complete VBN theory to provide a holistic view of how the variables of both theories work, notably in high-cost RCB context (e.g. the RACB of present study).The findings of Ates (2020) revealed that integrating TPB variables with morality variables accounts for more explained variance in pro-environmental behaviour (R2=0.488) compared to the use of individual TPB (R2=0.464). Ates (2020) highlighted that other important variables (e.g., egoistic value, altruistic value, new ecological paradigm belief) presented in the VBN should be taken into consideration. However, Ates’ (2020) study is confined to general RCB, not specific behaviour like green purchase behaviour. Another study by Li et al. (2021) combined the TPB, the diffusion of innovation theory (DOI), and the personal norm variable into a conceptual framework. However, the integration of the whole VBN variables were excluded from their study. 

Although the previous literature highlighted that habits should be included in studies for a more accurate prediction of repetitive behaviour, this factor had been ignored in the TPB and VBN models (Jansson et al., 2010; Verplanken & Aarts, 1999; Cheung et al., 1999). In response, extensive research added habits into the TPB model. For instance, Cheung et al. (1999) revealed that the explained variances of the TPB increased from 32.5% to 67.4% after incorporating habits into the TPB model to explain wastepaper recycling behaviour. Besides, the impact of TPB variable i.e. SN on VBN factors (e.g. AR, and PN), and habits are under-explored (Khan et al., 2022). 

Moreover, most of the prior RCB studies that combined the TPB and the VBN theory were focused on general pro-environmental behaviour (Ates, 2020; Bamberg & Moser, 2007), or low-cost, pro-environmental behaviour in the choice of transportation (Heath & Gifford, 2002; Bamberg & Schmidt, 2003) and the purchase of organic products (Li et al., 2021). The present study considers high-cost pro-environmental behaviour in the purchase of green computers that are more expensive than conventional computers.

To address the above research gaps, the objective of the study is to investigate the antecedents of RACB by integrating the TPB and the VBN theory with the habits variable. The outcome of this research will provide new knowledge of how the theories can be integrated to explain RACB. With this understanding, recommendations can be provided to policymakers, NGOs, manufacturers, and marketers to promote sustainable consumption through RACB.”

Comment:

Methodology and result presentation: The process of data analysis in this paper is too simple and rough. The author needs to present each step of data analysis in the concise way. Also, the title of table 4 need to be revised. The author didn’t present the direct and indirect effects clearly. Please enhance the mentioned parts.

Response: 

Thank you for your feedback. 

We are following the two-stages approach proposed by Anderson and Gerbing (1988) for SEM analysis. We added the following in Section 3.6: 

“As suggested by Anderson and Gerbing (1988), the two-step approach was used in this study. In the first step, the measurement model was analysed for adequacy, and confirmatory factor analysis (CFA) was used to test the validity and reliability of the measurement model. The second step involved testing the structural model and hypotheses through assessing the path coefficients for each hypothesised relationship.”

The title of table 4 has been revised. We only test the direct effect. 

Comment:

The integration of the TPB and VBN models cannot be seen as the theoretical contribution. Without having enough implications/contributions associating with your core concepts (TPB or VBN), the value of the work will be much weakened. Please enhance your implications/contributions parts in resubmission.

Response: 

Thank you for your feedback. 

The theoretical contribution is not just the integration of the TPB and VBN. In this study, we included the complete model for both the TPB and VBN theory and added the habit variable and the research was conducted in the context of high-cost purchase decisions e.g. acquisition of individual computers in emerging economy (i.e. Malaysia). This study unveils the complex relationships among the two models and the habit variable. Thus, this contribution is unique.

The following paragraphs have been rewritten to strengthen the theoretical implications (section 5.5) of this study:

“This study enriches the literature about responsible purchasing behaviour by addressing the absence of empirical studies examining the level of RCB and the antecedents of adopting RCB in the context of high-cost purchase decisions related to the acquisition of computers by individual consumers, particularly in emerging economy (i.e. Malaysia) comprised of low- and middle-income groups, fast-growing computer penetration (Worldometers, n.d.), and collectivist cultures. 

 This study provides holistic views relating to the integration of the complete TPB and VBN models with the habits variable in the high-cost pro-environmental behaviour domain, filling the literature gap where most of the scholars either dived into individual models like TPB (e.g., Lago, 2020; Kang & Moreno, 2020) or VBN (e.g., Sharma and Gupta, 2020), or extended individual models with additional variables (e.g., Ates, 2021). It unveils the complex relationships among the two models and the habits variable. Thus, this contribution is unique.

 The present findings confirm that a cognitive deliberative process occurs and that self-interest variables are significant when environmentally responsible behaviour is costly. Also, biospheric values are the main trigger of the TPB’s self-interest variable and VBN’s morality variable in forming decisions to adopt RACB. This further re-affirms that biospheric values are distinct from altruistic values in the context of pro-environmental behaviour (Nguyen, 2016; Ates, 2021). SN are a direct predictor of RACB (TPB variable), PN (VBN variable), AR (VBN variable), and HA (TIB). This further substantiates that SN are a particularly important element in the context of developing countries with collectivist cultures. In addition, HA affects intention but not RACB directly. This implies that individuals’ decisions will be affected by their HA in reasoning and assessing the costs and benefits of each alternative (i.e., RACBI) in the context of high-cost pro-environmental behaviour.”

Comment:

There is a lot of inappropriate/non-typical English/nonsense expressions, the language is not academic-oriented enough. Some tables in the paper should be formatted consistently with three-line format. Before re-submission, I strongly suggest the author need to send the draft to proofreader/copy-editor for improvement.

Response:

 Thanks for the comments. We corrected the table and sent the manuscripts for proofreading.

---

## [Decision Letter · Decision Letter 2]

7 May 2023

Antecedents of the responsible acquisition of computers behaviour: Integrating the theory of planned behaviour with the value-belief-norm theory and the habits variable

PONE-D-22-19976R2

Dear Yuen Yee Yen,

We’re pleased to inform you that your manuscript has been judged scientifically suitable for publication and will be formally accepted for publication once it meets all outstanding technical requirements.

Kind regards,

Tai Ming Wut

Academic Editor

PLOS ONE

Additional Editor Comments (optional):

Reviewers' comments:

Reviewer's Responses to Questions

**Comments to the Author**

1. If the authors have adequately addressed your comments raised in a previous round of review and you feel that this manuscript is now acceptable for publication, you may indicate that here to bypass the “Comments to the Author” section, enter your conflict of interest statement in the “Confidential to Editor” section, and submit your "Accept" recommendation.

Reviewer #4: All comments have been addressed

2. Is the manuscript technically sound, and do the data support the conclusions?

Reviewer #4: Yes

3. Has the statistical analysis been performed appropriately and rigorously? 

Reviewer #4: Yes

4. Have the authors made all data underlying the findings in their manuscript fully available?

Reviewer #4: Yes

5. Is the manuscript presented in an intelligible fashion and written in standard English?

Reviewer #4: Yes

6. Review Comments to the Author

Reviewer #4: Very clear results and discussions.

The contributions are clear.

I recommend that this paper be accepted for publication in the journal.

7. PLOS authors have the option to publish the peer review history of their article (what does this mean?). If published, this will include your full peer review and any attached files.

Reviewer #4: No

---

## [Editor Report · Acceptance letter]

22 May 2023

PONE-D-22-19976R2 

Antecedents of the responsible acquisition of computers behaviour: Integrating the theory of planned behaviour with the value-belief-norm theory and the habits variable 

Dear Dr. Yee Yen:

I'm pleased to inform you that your manuscript has been deemed suitable for publication in PLOS ONE. Congratulations! Your manuscript is now with our production department. 

Kind regards, 

on behalf of

Dr. Tai Ming Wut 

Academic Editor

PLOS ONE